# Attention expedites target selection by prioritizing the neural processing of distractor features

Mandy V. Bartsch [1,2✉], Christian Merkel[2], Mircea A. Schoenfeld[1,2,3] & Jens-Max Hopf [1,2✉]

Whether doing the shopping, or driving the car – to navigate daily life, our brain has to rapidly identify relevant color signals among distracting ones. Despite a wealth of research, how color attention is dynamically adjusted is little understood. Previous studies suggest that the speed of feature attention depends on the time it takes to enhance the neural gain of cortical units tuned to the attended feature. To test this idea, we had human participants switch their attention on the fly between unpredicted target color alternatives, while recording the electromagnetic brain response to probes matching the target, a non-target, or a distracting alternative target color. Paradoxically, we observed a temporally prioritized processing of distractor colors. A larger neural modulation for the distractor followed by its stronger attenuation expedited target identification. Our results suggest that dynamic adjustments of feature attention involve the temporally prioritized processing and elimination of distracting feature representations.

[1] Leibniz Institute for Neurobiology, Magdeburg, Germany. [2] Department of Neurology, Otto-von-Guericke University, Magdeburg, Germany. [3] Kliniken Schmieder Heidelberg, Heidelberg, Germany. ✉email: mandyvbartsch@gmail.com; jens-max.hopf@med.ovgu.de

When searching for tomatoes in a crowded veggie counter, one will most likely rely on their red color to spot them. That is because our brain can easily select a specific color among competing distractor color signals (green cucumbers, orange carrots, etc.), and guide our attention to locations of its occurrence[1,2]. Such guidance by color is particularly efficient, since attention to color can operate in parallel across the entire visual field, irrespective of item locations, thereby allowing for a rapid localization of colored objects. The location-independent nature of such feature selection processes is referred to as spatially global feature-based attention (GFBA), and its underlying neural correlates have been well-characterized both in the human[3–11], and the monkey[12–17]. At the single-neuron level, GFBA is assumed to arise from a multiplicative gain enhancement of feature-selective units in the visual cortex tuned to the attended feature value[13,15,17]. Consistently, EEG/MEG experiments in humans revealed that GFBA is associated with gain enhancements of the neural population response for the attended feature, starting around 150–200 ms after stimulation onset in extrastriate visual cortex areas[5,6,18–21]. It is reasonable to assume that efficient target identification will depend on how rapidly this global biasing can be built up for the attended feature. Still, the processes that serve to adjust feature selectivity dynamically among multiple attended feature values are little understood.

In most GFBA experiments, the target is defined by a single constant feature value (e.g., one color) for a complete experimental trial block, such that participants implement a stable preset bias for that specific target feature value. In everyday life, however, target items appear in different shades, and we often look for several things simultaneously. For example, with both red tomatoes and green avocados being on our shopping list, we often do not know in which order we will find those items. As a consequence, we are required to hold a parallel bias for both colors (attentional template for red and green), but to adapt (bias) color selectivity 'on the fly' to the color of the vegetable that happens to be encountered in a given moment. That is, when first coming across tomatoes, our brain should strengthen the color bias for red, while attenuating the response to green, a currently distracting target color alternative (see Fig. 1). But how does the brain quickly shift color selectivity between colors when not knowing before which of them will be target-defining and which will be distracting?

The aim of the present study is therefore to investigate the cortical dynamics of adjusting color selectivity among two target colors in the moment one or the other color has to be selected. To accomplish this, we employ a modified version of the unattended probe paradigm (UPP) used in Bartsch et al.[5] experiment 3, with the task requiring a color discrimination among two target color values. We informed participants that the upcoming target will be drawn in one of two possible target colors (e.g., red or green). A simultaneously presented, unattended color probe could be drawn in either of those colors. Analyzing the brain response to the probe allowed us to assess the temporal evolution of the GFBA response as a function of whether the color must be biased for target identification (probe matches present target color, PC), or de-emphasized in favor of the presented target color (probe contains distracting target color alternative, DC) on a given trial. We also add a control condition, where subjects view exactly the same stimuli, but are asked to discriminate the orientation of the target, while color is completely task-irrelevant. On each trial, there is a 50% chance that one or the other target color appears. Hence, it would be a reasonable assumption that participants implement a balanced top-down bias (attentional template) for both target colors. In the moment the target appears, however, the bias must shift to the color value of the currently presented target.

Regarding the time course and amplitude of the GFBA response of the two colors, several scenarios are possible, which are illustrated in Fig. 1c. There may be an initial GFBA response for both the present (PC, red) and the distracting (DC, green) target color appearing with the same temporal onset. The DC, however, may raise to a smaller amplitude and soon fade away as the neural bias for this color declines in favor of the PC (green solid), fitting previous observations in Bartsch et al.[5] A related possibility is that the DC does not only fade but will be actively suppressed below the level of an unbiased color (green thick dashed). As an extreme, this suppression of the DC could already start at the onset of the GFBA response (green thin dashed).

## Results

**Dynamics of color attention during color and orientation discrimination.** Twenty-two human adult volunteers participated in both the color and the orientation task (see Fig. 2 for experimental design). They were required to attend to a semicircle presented in the left visual field (VF) and to either identify its color (color task), or orientation (orientation task). Importantly, the target was always randomly drawn in one of two possible target color alternatives (e.g., red or green), such that after stimulus onset, participants had to quickly adapt their bias 'on the fly' toward the present target color (PC), and away from the distracting target color alternative (DC). The temporal development of color selectivity was tracked by recording the event-related potential (ERP) to simultaneously presented irrelevant color probes (unattended probe paradigm, UPP, for details see Methods). This resulted in three different trial types per experimental condition, i.e., the probe could contain the PC, the DC, or a non-target color. The response difference measured between attended colors (PC and DC) and irrelevant non-target colors was taken as global feature-based attention effect (GFBA) and served to track cortical color biasing. When participants performed the color task, we expected to observe a dynamical adjustment of attentional color selectivity in favor of the PC (see Fig. 1c). The orientation task served as an experimental control condition rendering the color of the current target irrelevant. That way, we could distinguish between the effects related to color attention (color task) and effects that might arise when discriminating the stimuli without explicitly attending to their color (orientation task).

**Behavioral results—distracting color alternative impairs performance in the color attention task.** Figure 2c, d displays response time (RT) and response accuracy for the different trial types. As can be seen, participants responded fast (<410 ms) and with high accuracy (>92% correct) across all conditions. However, responses seemed to be slightly slower and less accurate when performing the color task, which was most obvious when probes contained the distracting target color alternative (DC, gray bars). Conducting 2×3 rANOVAs with the factors TASK (orientation/color) and COLOR (PC/DC/non-target) revealed significant main effects for TASK (accuracy: $F[1,21]=5.15$, $p = 0.034$, RT: $F[1,21]=11.17$, $p = 0.003$), confirming the performance decrement in the color task. As expected, there was also a main effect of COLOR (accuracy: $F[2,42] = 5.08$, $p = 0.011$; RT: $F[2,42] = 7.56$, $p = 0.002$) that showed a significant interaction with TASK for response time ($F[2,42] = 9.44$, $p = 0.002$) but not response accuracy ($F[2,42] = 2.60$, $p = 0.101$). Subsequent t-tests confirmed that, for the color task, responses were significantly slower and less accurate when probes matched the DC compared to when probes matched the PC (accuracy: $p = 0.002$; RT: $p = 0.001$), or a non-target color (accuracy: $p = 0.049$; RT: 0.009) with no difference between the latter (accuracy: $p = 0.108$; RT: $p =$

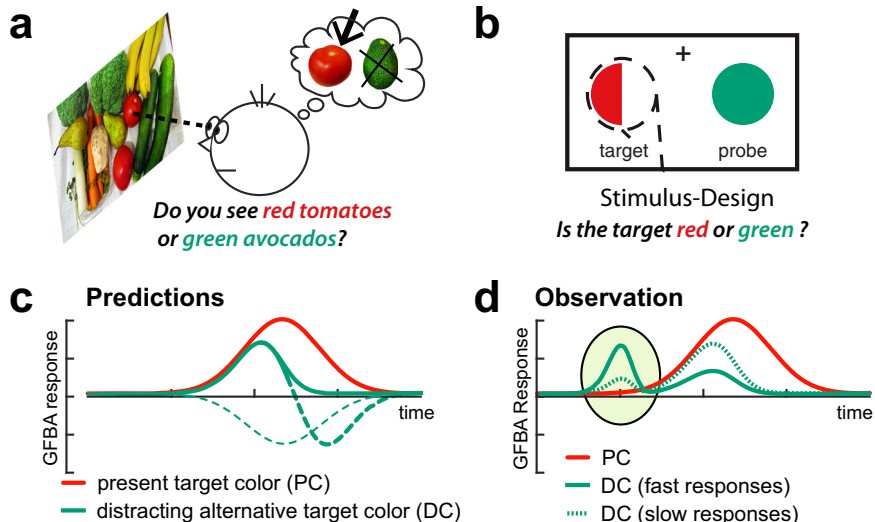

**Fig. 1 Dynamics of attentional color biasing. a** Motivation. When searching for both red and green items, not knowing what we will encounter first, our brain must decide 'on the fly' which color is currently contained in a target object (here: red) and which color would be rather distracting (here: green), and adjust the color bias in the brain accordingly. **b** Experimental idea. To investigate this color biasing dynamic independent of other influences like object location, we created simplified stimuli where the target location was fixed, but its color changed unpredictably between two colors (see Fig. 2 for details). **c** Predictions. The color selection bias in the brain was assessed as the amplitude of the global feature-based attention (GFBA) response to that color (for details see Methods). Participants may initially bias both possible target colors (here: red and green). The response to the distracting color alternative (DC, here: green), will then decay (green solid) as the neural bias for this color declines in favor of the present target color (PC). Alternatively, the DC might become actively suppressed below baseline either with a delay (green thick dashed), or right from the beginning of the GFBA modulation (green thin dashed). **d** Observation. Contrary to our predictions, the processing of the distracting target color alternative (DC, green) gained temporal priority (marked by the ellipse). On trials with a fast response time—i.e., fast identification of the target's color—participants showed a prominent early selection of the DC followed by its stronger attenuation in the time range of maximal biasing of the present target color (PC, red). For slow responses (green dashed), the early response to the DC and its subsequent attenuation were less pronounced.

0.056). For the orientation task, we did not observe significant influences of the probe color on performance besides a slight response time increase when the probe contained the PC compared to a non-target color ($p = 0.016$, all other $p > 0.17$). That is, when asked to discriminate the target's orientation but not its color, a probe matching its color might have been slightly distracting.

Taken together, though behavioral performance was very high in both tasks, when participants performed the color task, the DC significantly slowed performance, indicating that presenting the target color alternative did, indeed, distract the processing of the present target color, which slightly delayed responses. Consistently, no such DC performance decrement was observed under conditions of the orientation task, indicating that participants successfully implemented different attentional task sets with the alternative color of the target being irrelevant and, hence, not distracting during attention to orientation.

**Event-related potential responses**. According to previous work[5,6,18,19], we expected to find an increased negativity for probes containing attended compared to unattended colors at parieto-occipital electrode sites contralateral to the probe location (here: signal averaged across PO3 and PO7). Specifically, when subjects performed the color task, we should observe modulations for both target color alternatives (PC and DC are both part of the attentional set). In contrast, when discriminating the orientation of the target irrespective of its color, color biasing should, if at all, only be present for the PC (i.e., for the color contained in the object under discrimination). An overall sliding window $2 \times 3$ ANOVA (see Methods, Statistical validation of amplitude differences) with the factors TASK (color/orientation) and COLOR (PC/DC/non-target) revealed a significant early TASKxCOLOR interaction (73–96 ms) and a late main effect for COLOR (167–254 ms). Subsequent analyses focusing

on those time ranges revealed pronounced early and late processes of color biasing for the color task, and a more general task-independent enhancement of the PC in the later time range, as detailed in the following.

**Color task—early cortical modulation for the distracting color**. Figure 3a shows the ERP responses for the color task. When participants had to decide which of two colors was present in the focus of attention, we unexpectedly find that in comparison to the non-target color (black dashed line, reference condition), it is the distracting color alternative (DC, gray line), but not the present target color (PC, black solid line) that elicited the earliest negative deflection (73–96 ms). This early negative deflection is then followed by a later negativity (167–254 ms). The latter is elicited by both target color alternatives, although it is most prominent for the present target color (PC). To better visualize the GFBA effects, the response to the unattended non-target color was subtracted from that of the DC (Fig. 3b) and the PC (Fig. 3c). A statistical analysis confirmed that the early color effect was only present for the DC ($p = 0.0179$) but not for the PC ($p = 0.6521$). The late effect, however, was apparent for both DC ($p = 0.0403$) and PC ($p < 0.0005$), albeit much more pronounced in the latter (mean amplitudes differ significantly, $p = 0.0103$). The respective topographic field distributions of the GFBA responses reveal similar parieto-occipital distribution maxima at electrodes PO3 and PO7 for the early and late effect (Fig. 3b, c, on the right). Although there seems to be a small early modulation for the PC outside the time range of investigation (Fig. 3c), an additional explorative sliding window $t$-test (sample-by-sample, 11.8-ms window) between 0 and 130 ms revealed no significant modulation before or shortly after the early time window (all $p > 0.05$).

The modulation observed in the N1/N2 time range (here referred to as "late" effect) is well in line with previous literature,

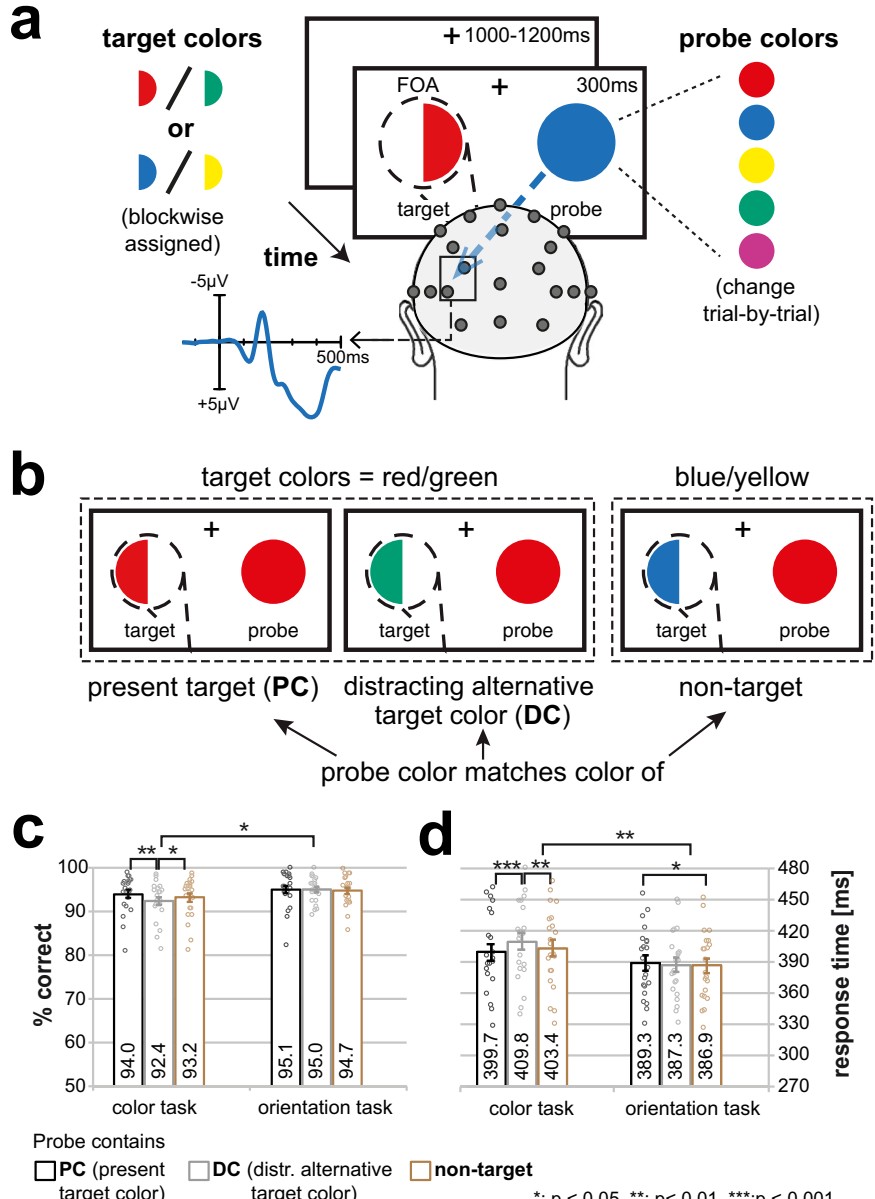

**Fig. 2 Experimental design and behavioral results. a** Participants attended to a colored hemicircle presented in the left VF (dashed line = spatial focus of attention, FOA) and reported by button press either its color (color task blocks) or the side (left/right) of its convexity (orientation task blocks). The target varied trial-by-trial unpredictably between the two blockwise-assigned target colors (i.e., between red and green, or between blue and yellow). On each trial, the color probe simultaneously presented in the right VF was randomly drawn from five colors (red, green, blue, yellow, and magenta). The effects of global feature-based attention (GFBA)—as a measure for attentional color selectivity in the visual cortex—were assessed by comparing the event-related brain response elicited by an unattended color probe as a function of whether it matched the present target (PC), the distracting alternative target color (DC), or neither of them (non-target). **b** Trial types. The probe (here: red) could either contain the PC, the DC (here: red probe but green target in an attend red/green block), or could represent a non-target color currently not relevant (here: red probe in an attend blue/yellow block). Behavioral performance. Shown are the percentage of correct responses **c** and response times **d** of both the color and the orientation task for all trial types. Participants ($n = 22$) responded highly accurately and fast across all conditions. However, the performance was slightly lower in the color task, most prominent as a response delay on trials where the probe matched the distracting alternative target color (DC, gray bars). The error bars represent the standard error of the mean (SEM). Black, gray and brown dots represent data points of individual participants.

where such negative amplitude modulations have been observed for the attended color[5,18,19]. Those N1/N2 effects were found to reflect top-down modulations propagating in reverse-hierarchical direction in extrastriate visual cortex. Also fitting previous observations of Bartsch et al.[5] (experiment 3), this effect is smaller and more transient for a color that is part of the attentional set but not discriminated on a given trial (here: DC). The response in the N1/N2 time range therefore matches the

prediction illustrated in Fig. 1c (red and green solid), with an initial rise of the response for both attended colors, but a smaller amplitude for the DC, which is never attenuated below the level of the non-target color responses. To our surprise, and not fitting any of our predictions, those late modulations were preceded by a very early negativity around 70–100 ms relative to the non-target color, which appeared only for the DC but not the PC. This seems to be counterintuitive as it suggests that the DC gains an early

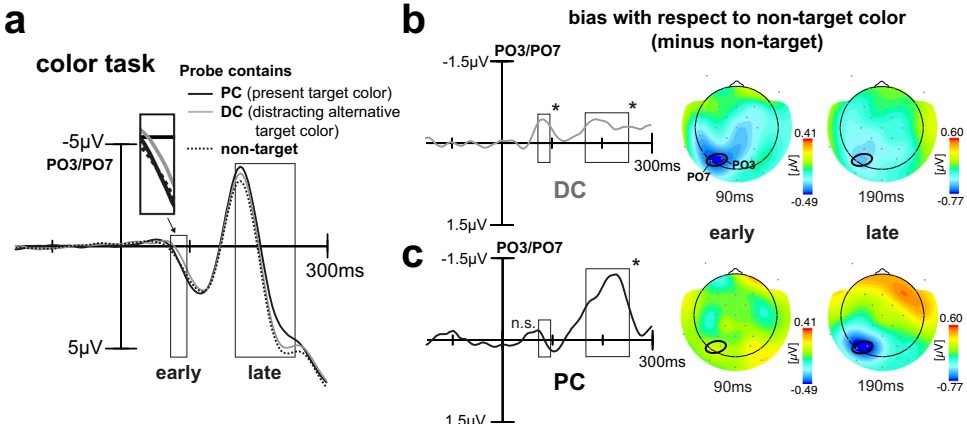

**Fig. 3 ERP results for the color task. a** Shown is the ERP elicited by the probe at PO3/PO7 (signal averaged) for the different trial types when participants (*n* = 22, signal averaged) were to discriminate the color of the target. Rectangles highlight time ranges of significant brain response variations as derived by the 2×3 rANOVA. Surprisingly, participants show a pretty early modulation (higher relative negativity around 73–96 ms) for the DC (gray line), see inset for an amplified depiction. Significant modulation for the color of the PC emerges later (167–254 ms). Difference waveforms for DC minus non-target color (**b**) and PC minus non-target color (**c**). The respective topographical field maps on the right display representative time points at early and late modulation maxima, positions of electrodes used for analyses are highlighted (black ellipses). In the early time range, there is a prominent modulation for the DC but not the PC. In the late time range, this pattern becomes inverted with a strong modulation for the PC and a comparably small modulation for the DC.

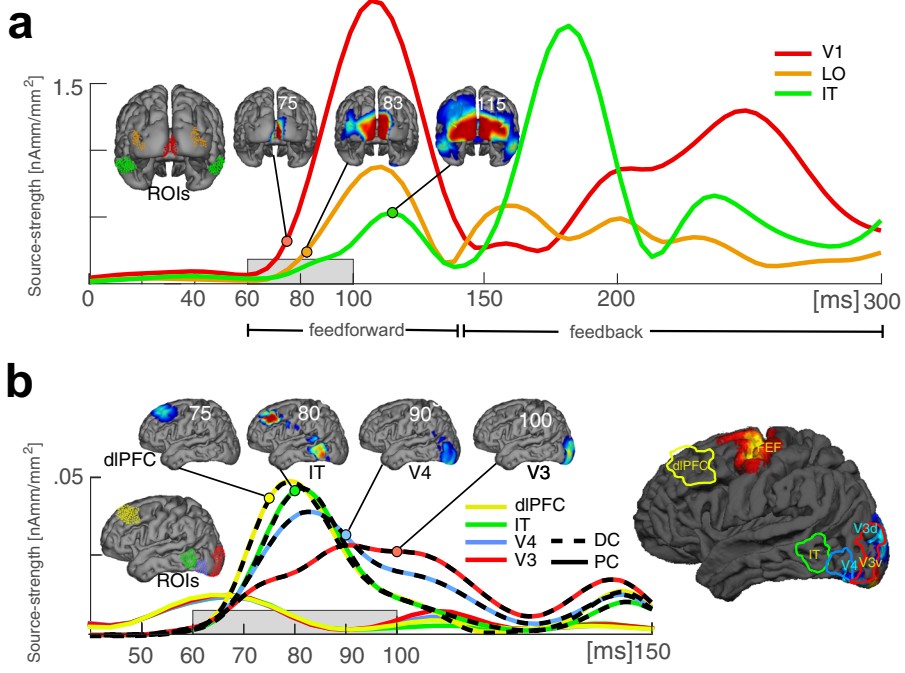

**Fig. 4 Time course of cortical current source activity for the color task. a** Shown is the propagation of stimulus-elicited activity in the visual cortex after stimulus onset (current source activity of the average across target (PC), non-target and distractor (DC) color probes; signal averaged across participants (*n* = 22)). The waveforms show time courses of source strength at selected ROIs (red: primary visual cortex, orange: lateral occipital (LO) cortex, and green: IT cortex) (see Methods). Small 3D views show current source-density maps at selected time points illustrating the initial feedforward sweep of processing. **b** Time course of the source activity underlying the very early distractor selection (DC minus non-target, color dashed, shown between 40 and 150 ms) at selected ROIs depicted in middle-sized and large 3D view (yellow: dorsolateral prefrontal cortex, green: IT cortex, blue: V4, and red: V3). For target-colored probes, the time course of source activity at the respective ROIs does not show comparable modulations (PC minus non-target, color solid). Small 3D views show source-density maps at selected time points illustrating the prefrontal-to-ventral extrastriate propagation of the DC biasing. As can be seen, the very early selection bias for the distracting color appears already during the initial feedforward sweep of information processing (compare time range highlighted by gray horizontal rectangles in **a** and **b**).

selection bias above the PC, and that this bias arises already during the initial feedforward sweep of processing in the visual cortex. To verify the latter, we analyzed the time course of cortical current source activity underlying the early DC negativity to compare it with the initial forward propagation of stimulus-elicited activity in the visual cortex (Fig. 4a). As expected, the stimulus-driven visual response arises first in the primary visual cortex (V1, red ROI and source wave) around 65 ms, followed by source activity starting between 70 and 80 ms in lateral occipital regions (orange), consistent with mid-level dorsal stream area

hMT. With a further delay, source activity arrives in anterior ventral extrastriate cortex (IT, maximum at 115 ms) where it disappears ~140 ms. This marks the end of the initial feedforward sweep of processing and the beginning of the canonical feedback activity starting in IT (green maximum at ~180 ms) and reaching early the visual cortex (V1) with a further delay (maximum at ~250 ms). Importantly, current source activity underlying the early negativity for the DC (Fig. 4b, color dashed) arises during the initial feedforward sweep with almost no delay relative to V1 (70 ms) in a region of the dorsolateral prefrontal cortex (dlPFC) anterior to FEF (frontal eye field) (yellow ROI and yellow dashed source wave) roughly consistent with Brodmann areas (BA) 46/9[22,23]. This frontal activity is rapidly followed by source activity in anterior lateral extrastriate cortex (IT) around 80 ms (green dashed), and in more posterior lateral areas V4/V3 around 90–100 ms (blue/red dashed). Hence, the early DC biasing is generated by a sequence of prefrontal-to-ventral extrastriate source activity, which clearly arises during the initial feedforward sweep of processing in the visual cortex. Importantly, when analyzing the source activity in the same ROIs for the PC, this early sequence of source activity in prefrontal-to-ventral extrastriate cortex was absent (Fig. 4b, color solid). Specifically, the PC source activity showed only a weak early peak that was simultaneously present in all ROIs, presumably reflecting noise fluctuations.

**Stronger early cortical modulation for the distracting color expedites target selection.** Given the 50% chance that the one or the other target color appears on a given trial, one would expect that the top–down bias for both colors is overall balanced, and that upon stimulus onset, the neural processes mediating the selection of the PC are involved as fast as possible. The observed response pattern, instead, suggests that the DC undergoes prioritized processing. One possibility would be that the DC is selected with temporal priority in order to rapidly build a representation that serves its rejection from further processing (selection for rejection). Alternatively, the early DC modulation may reflect an immediate attenuation to facilitate later rejection. In both cases, the prioritized DC processing would ultimately facilitate the selection of the PC. If this is the case, the amplitude of the early negativity would be expected to inversely relate to the time it takes to discriminate the PC. Furthermore, if the early negativity reflects the selection for rejection of the DC from current processing, a higher amplitude of the early negativity should be associated with a smaller late GFBA response to this color. To address this possibility, we compared the GFBA response to the PC and DC after separating the data into fast and slow correct responses (Median response time-split analysis, see Methods). Figure 5 shows the respective brain responses for the color task of trials where subjects gave fast versus slow responses.

As can be seen in Fig. 5b, c, the amplitude of the early DC modulation did, indeed, vary with response time as predicted. The DC minus non-target difference waveform shows a significant early modulation for fast ($p = 0.0019$, gray line in Fig. 5b) but not for slow ($p = 0.3755$) DC trials (gray line in 5c). In contrast, in this early time range, no response difference is seen between fast and slow PC trials (black traces in 5b, c). Furthermore, while there is a small GFBA response to fast DC trials in the late time range that onsets with the GFBA response to fast PC trials, it is rapidly attenuated (Fig. 5b). This is in contrast to slow trials (Fig. 5c), where the late GFBA response to DC trials is not attenuated. Note, the early modulation in DC trials is present in fast but effectively absent in slow trials. A further RT-quartile split analysis of the data shown in Supplementary Figure 1 qualifies this observation. The early

modulation effect is gradually reduced in amplitude with increasing response time, with a large-amplitude effect appearing in very fast (1st quartile) and a smaller effect in fast trials (2nd quartile). The early DC modulation is minimal in slow (3rd quartile) and absent in very slow trials (4th quartile), indicating that it is primarily driven by fast target discriminations.

The bar graphs on the right of Fig. 5b, c show a direct comparison of mean amplitude values in the early and late time range of the RT median split data. On fast DC responses (gray bars, 5b), the early modulation was significantly higher ($p = 0.0132$), while the late selection was reduced relative to slow (gray bars, 5c) responses (marginally significant, $p = 0.0558$). A current source analysis of the DC-related modulation differences (fast minus slow responses at 70–95 ms, Fig. 5d) shows that the performance increment due to prioritized DC selection is primarily caused by differences in the early visual response, either reflecting an initially stronger sensory representation of this color or an inhibitory signal for attenuation in early and mid-level visual cortex during the feedforward sweep of processing.

A rANOVA on ERP amplitudes with the main effects EARLYLATE (early GFBA amplitude, late GFBA amplitude) and FASTSLOW (fast responses, slow responses) confirmed a significant EARLYLATExFASTSLOW interaction ($F[1,21] = 18.15$, $p = 0.00035$) with no significant main effects (EARLYLATE: $F[1,21] = 0.164$, $p = 0.689$; FASTSLOW: $F[1,21] = 0.001$, $p = 0.982$) for the DC. For the PC, on the other hand, there was no significant early modulation at all but a strong late bias that did not vary with response time. As expected, the analogous rANOVA found a significant main effect for EARLYLATE ($F[1,21] = 33.13$, $p < 0.0005$) but none for FASTSLOW ($F[1,21] = 0.004$, $p = 0.953$), or the interaction ($F[1,21] = 0.37$, $p = 0.547$). Importantly, with a mean fast response time of 347 ms on DC trials, responses were initiated well after (and not before) the early modulation, probably during or shortly after the late time range, in line with an influence of the early cortical distractor processing on the speed of target identification.

Together, the RT split pattern clearly supports the idea that a preferential processing of the DC and its subsequent rejection expedites the identification of the target's color. This conclusion implies that it is the prominence of the early sensory representation of the DC and not so much the very processing of the PC itself that is important for efficient target identification. This leads to a notable prediction: when one would strengthen the cortical representation of the PC, a larger early modulation for the DC would be required to instantiate a prioritized representation of the DC against this stronger PC bias. Alternatively, the early distractor prioritization may not be mandatory for task performance. A stronger representation of the target might simply eliminate the need for the early DC processing (and later DC rejection). One way to assess the influence of the strength of the target representation would be to analyze the effects of intertrial color priming[24–29]. Specifically, repeating the target feature (PC) on subsequent trials should strengthen its sensory representation, leading to faster response times and decreased distractor interference.

To determine the influence of color priming, we split conditions into trials where the target was repeated and trials where the target color switched on subsequent trials (Fig. 6a, for details see Methods, Target repetition analysis). Participants responded about 40 ms faster on target repetition trials, irrespective of probe color (PC: 381 ms vs. 419 ms, DC: 390 ms vs. 428 ms, non-target: 387 ms vs. 419 ms). A two-way rANOVA with the factors COLOR (probe matches PC/DC/non-target) and REPETITION (target repeated/switched) revealed a significant main effect of REPETITION ($F[1,21] = 43.81$, $p < 0.0005$) and of COLOR ($F[2,42] = 9.32$, $p = 0.001$), but no interaction between

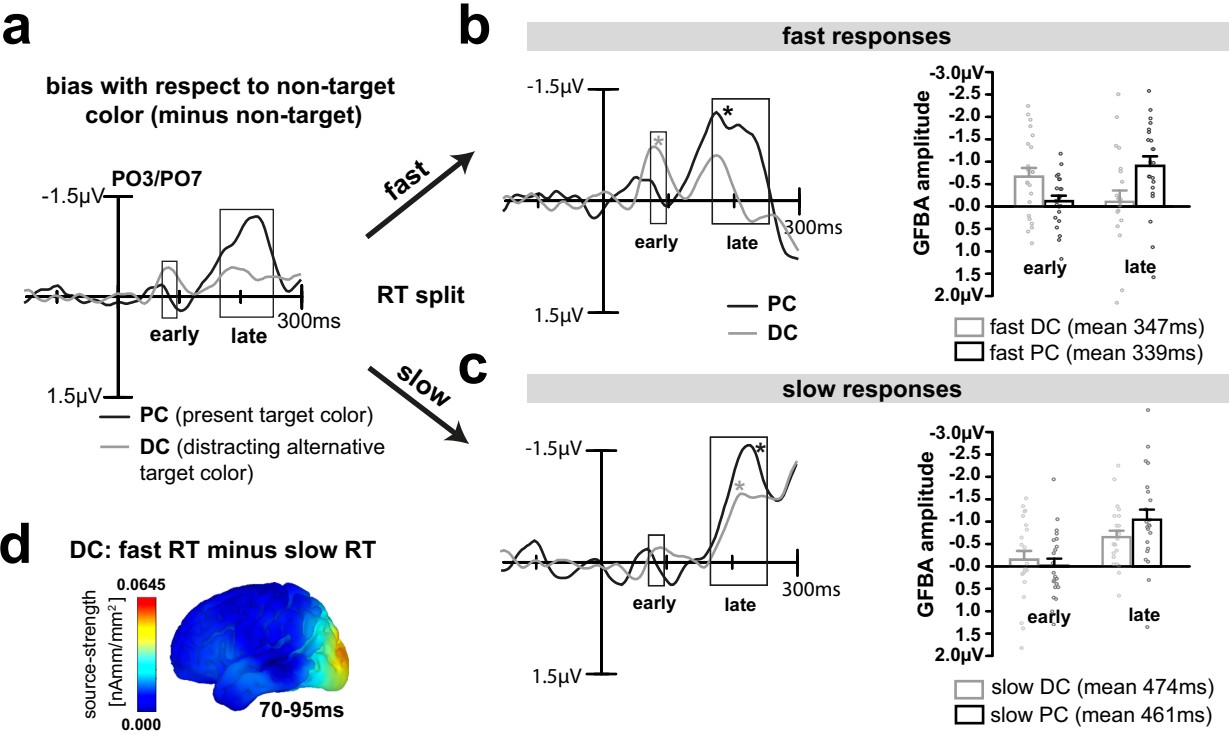

**Fig. 5 Median response time-split analysis. a** Difference waveforms for the PC (black) and DC (gray) replotted together from Fig. 3b, c for better comparison. Rectangles indicate previously determined early and late time ranges of significant experimental variation. Median split into fast (**b**) and slow (**c**) response times (RT) for DC (gray) and PC (black). ERPs: fast and slow minus non-target difference waveforms averaged across participants (*n* = 22). Stars indicate significant mean amplitude modulations in the early and late time ranges (*p* < 0.05). An explorative sample-by-sample sliding *t*-test (0–100 ms, 11.8-ms window) found no effect for fast PC trials within or before the early time window. Bar graphs: mean GFBA amplitudes of the early and late time range are shown for fast (**b**) and slow (**c**) responses. For DC, the early negativity was higher when participants responded fast compared to slow, which was inversely correlated to the size of the late bias (significant early/late GFBA amplitude × fast/slow RT interaction, *p* = 0.00035, see text for details). For the PC, in contrast, there was no significant difference in GFBA amplitudes between fast and slow responses but always a strong late bias. The error bars represent the standard error of the mean (SEM). Black and gray dots represent data points of individual participants. **d** 3D current source-density map for the early DC biasing (fast RT trials minus slow RT trials) between 70 and 95 ms (25-ms average). As can be seen, differences in source activity for fast and slow DC trials emerge in posterior extrastriate visual cortex (around V3).

them (F[2,42] = 1.02, *p* = 0.351). As visible in Fig. 6b, repeating the target color increased the early modulation and subsequent attenuation of the distracting color. Hence, the stronger neural representation of the PC on primed trials does not abolish but reinforce the early DC processing, underscoring its role for efficient target selection.

Taken together, the ERP responses observed during the color task indicate that a strong initial processing of the DC followed by a weaker late DC response entails faster correct responses, presumably because of diminished distractor color interference in the later time range of target discrimination (maximal PC response in Fig. 1d). The pattern is suggestive of a mechanism akin to "selection for rejection" previously observed for distracting color singletons in visual search displays[30].

**Orientation task—rendering target color irrelevant abolishes early color-biasing effects**. To draw strong conclusions, it is important to verify that the GFBA effects observed in the color task, indeed, reflect pure modulations of attention to color and are not based on low-level sensory processes that differ among the experimental conditions. In particular, trials with the probe matching the PC versus the DC differ as to the diversity of color in the stimulus array. On PC trials, the same color appears in the left and right VF, while on DC trials different colors appear (see Fig. 2b). It is therefore critical to rule out that differences in the early modulation time range are triggered by such color

imbalance between VFs. To this end, we analyzed the brain responses to the very same physical stimuli when subjects were asked to discriminate the orientation of the target item, while color was completely task-irrelevant. Here, we expected no early modulation for the alternative target color (DC). However, as observed in previous work[5], we anticipated some later GFBA effect when probing the color of the target under discrimination (PC). Figure 7a shows brain responses averaged over trials with the same color assignment as in the color task, but when participants performed the orientation discrimination with color being irrelevant. As can be seen, there was no obvious early color effect, especially no higher negativity for the DC (gray line). Instead, the difference waveforms in Fig. 7b, c reveal a rather small countermodulation (positivity) in the early time range for both PC and DC which is, however, only significant for the DC (*p* = 0.0242). As expected, there was some late modulation for the PC in the late N1/N2 time range (7c, black line), but no effect for the DC (7b, late time window, gray line). The late PC modulation is, however, smaller compared to that of the color task (cf. Figure 3c), which is confirmed by a paired t-test (mean amplitudes differ, *p* = 0.0031). The late negativity in the orientation task is preceded by a smaller negativity around 130 ms. An explorative sliding window *t*-test in the time range between the early and late time window (sample-by-sample, 11.8-ms window), however, failed to reach significance. Importantly, the DC modulation pattern (early selection and subsequent attenuation) is eliminated

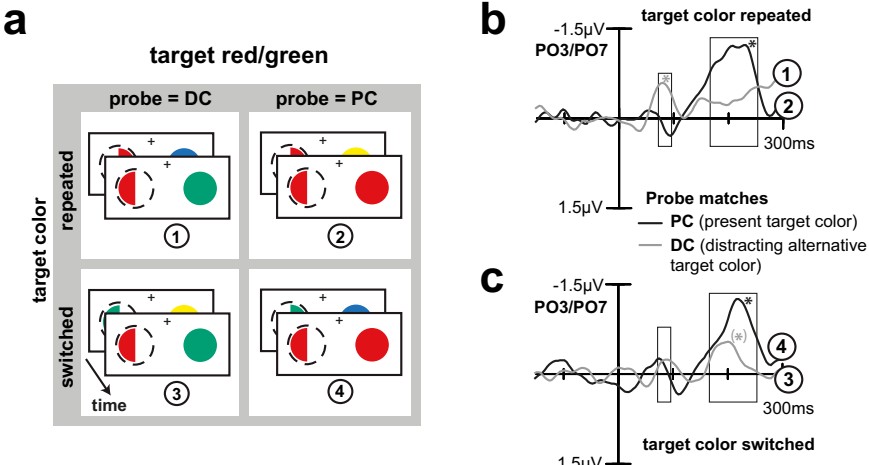

**Fig. 6 Target repetition analysis. a** Trials in which the probe contained the DC, or the PC were sorted according to whether the color of the previous target was repeated (stronger priming-driven color bias for the PC), or switched (weaker priming-driven color bias for the PC). Here exemplarily shown for trials of an attend red/green block and a red target. Difference waveforms (minus non-target) for PC (black) and DC (gray) probes for repeat (**b**) and switch (**c**) trials, signal averaged across participants ($n = 22$). Rectangles indicate previously determined time ranges of early and late color biasing. A stronger priming-driven bias for the PC (repeat trials) entailed a pronounced early modulation for the DC ($p = 0.0130$), which was much smaller and not statistically significant on switch trials ($p = 0.2761$). Corroborating the response time split (see Fig. 5), a strong early processing of the DC was followed by its weaker selection in the late GFBA time range and linked to faster target identification. Stars indicate significant mean amplitude modulations in the indicated time ranges ($p < 0.05$), the late GFBA response for DC under switch conditions (c, gray line) was only significant when considering a shorter time window (i.e., from 167 to 210 ms).

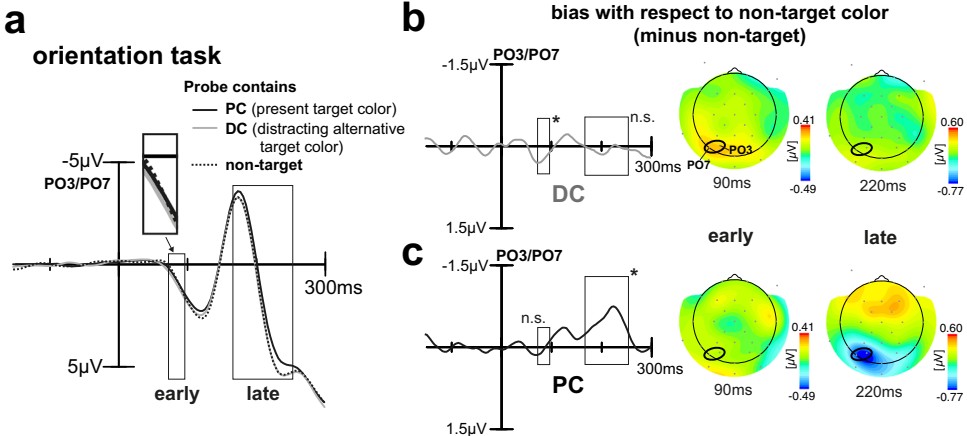

**Fig. 7 ERP results for the orientation task. a** Shown is the ERP elicited by the probe at PO3/PO7 (signal averaged) for the different trial types when participants ($n = 22$, signal averaged) were to discriminate the orientation of the target. Rectangles highlight time ranges of significant brain response variations as previously derived by the $2 \times 3$ rANOVA. This time, participants show no relative negativity for the DC (gray line), see inset for an amplified depiction. Again, there is a late negativity for the color of the present target object (PC) (167–254 ms). Difference waveforms for DC minus non-target (**b**) and PC minus non-target (**c**). The respective topographical field maps on the right display representative time points at early and late modulation maxima with the positions of the electrodes used for the analyses being highlighted (black ellipses). The difference waveforms reveal a small positive modulation for color in the early time range and a late enhancement (higher negativity in the N1/N2 time range) that is present for the color of the object being under discrimination only.

when participants did not attend to color and thus highly unlikely to be caused by mere low-level sensory differences among trial types.

To sum up, when participants have to decide among two possible target color alternatives (color task), they appear to first implement a very early selection process (73–96 ms) for the potentially distracting target color alternative (DC). This selection process, which may represent a rapid enhancement or attenuation of neural activity coding the DC, entails a reduced late biasing for that color in the N1/N2 time range (167–254 ms), where previous work reported template matching and feature discrimination processes to occur cf.[5,6,18]. Contrary to any of our predictions, the

bias for the color currently present in the target object (PC) is built up more slowly resulting in a strong late selection (167–254 ms) that would also be present—though smaller in amplitude— when performing a mere orientation discrimination of the object.

## Discussion

The aim of this study was to investigate the temporal dynamics of biasing color selectivity in favor of the currently presented target color, when an upcoming target appeared unpredictably in one out of two possible colors. To this end, we compared the event-related brain response (ERP) to spatially unattended probes drawn in the target color alternatives, when participant had to

decide 'on the fly' on a given trial which of them was currently contained in the target (present target's color, PC), and which was rendered a now distracting color alternative (DC). We expected observers to establish an initially balanced top-down bias for both colors, with the response to the DC soon fading away or being even actively suppressed (Fig. 1c). ERP responses showed, indeed, that observers established a top-down attentional bias for both colors, which is in line with our previous observations[5,6] as well as with work showing that observers can establish attentional control settings for multiple colors at once[31–35], and that guidance by two target colors is possible in visual search[36,37]. In the present study, both the PC and the DC, elicited global feature-based attention (GFBA) modulations roughly at the same latency in the N1/N2 time range (167–254 ms, here referred to as late effect). In line with experiment 3 of Bartsch et al.[5], the response to the DC was smaller in amplitude and decayed faster (Fig. 5a, late time range), fitting the prediction depicted in Fig. 1c, solid lines.

However, as shown in Fig. 1d, beyond our predictions, we observed that the DC but not the PC elicited a prominent and rather early modulation (peaking around 90 ms) roughly ~80 ms prior to the onset of the GFBA effects in the N1/N2 time range (Fig. 3b). The polarity and scalp topography of this early modulation was similar to that of the subsequent GFBA effects, suggesting that it reflects some form of early global color selection already during the initial feedforward sweep of processing in the visual cortex, which is functionally related to the following feedback-dependent GFBA response. Indeed, a current source-density analysis of the early modulation revealed that it is generated by a rapid sequence of activations in frontal-to-extrastriate visual cortex within the time range in which the feedforward sweep of processing just reached higher-level extrastriate areas (IT) (see Fig. 4). At first glance, this observation is counterintuitive, as one would expect that the prioritized selection of the DC impedes and delays the discrimination of the PC. However, further analyses revealed the opposite effect—a stronger early DC processing was associated with a faster target identification. What then, is the function of this early modulation? One possible explanation would be that the priority processing of the distracting target color alternative serves to build a temporary representation for eliminating or attenuating the selection bias for this color (selection for rejection). Note, the present data cannot decide whether this priority processing reflects the enhancement or attenuation of underlying neuronal responses. In any case, it would reasonably be most effective before the 'regular' cortical processes (N1/N2 GFBA effects) dealing with the representation and discrimination of the color of the present target start to arise. A blocking of DC processing in the subsequent GFBA time range would effectively bias the competition toward the PC[38,39], thereby expediting target discrimination. The median RT-split analysis of the present data supports this interpretation. A stronger early DC modulation in fast RT trials was associated with a smaller later GFBA amplitude (see Fig. 5b, gray lines/bars, and Fig. 1d for a schematic depiction), suggesting that the GFBA response to the DC was, indeed, attenuated in the N1/N2 time range. Importantly, for the PC, no significant variation of the GFBA amplitude as a function of RT was seen, indicating that the processing of the DC but not that of the PC was linked to response time differences. Hence, the present data suggest that selecting among unpredictably changing target colors with equal top-down bias involves a temporarily prioritized representation of the currently distracting color. This early representation presumably serves to suppress later GFBA responses to this color, thereby ultimately facilitating the discrimination of the actual target color.

The analysis of trial-by-trial color priming in the present experiment supports this interpretation. If the early DC modulation reflects, indeed, an active process to highlight the distracting target color alternative, it should be influenced by the strength of the cortical representation of the present target color. A priming-driven bias toward the PC on target color repeat trials[24,25,27] led to a stronger early DC modulation (and subsequent attenuation) consistent with activity counteracting the enhanced sensory bias for the PC on repeat trials to instantiate the priority representation of the DC. Importantly, the early DC selection was not itself a mere artifact of DC color priming, since the DC was not systematically repeated on those trials in the unattended color probe (previous probe color random, cf. Figure 6a).

One possible interpretation worth considering is that the early DC-related modulation may represent an attenuation of a positive ERP deflection rather than an enhanced negativity. In fact, under certain experimental conditions, it was shown by Zhang and Luck[40] that GFBA can influence the feedforward sweep of processing in the visual cortex very early on, in form of a positive-going modulation in the P1 time range. Theoretically, the early DC modulation may reflect a reduction of such P1 response. However, the feature-based P1 modulation in Zhang and Luck[40] was only seen under conditions of color competition in the spatial focus of attention, but not when a single color was presented (experiment 2 in[40]). As the present experiment did not involve color competition in the focus of attention (a single color appeared in the focus of attention), such P1 effect would not be expected to appear. Furthermore, an interpretation in terms of an attenuated P1 response for the DC relative to the PC would have to reconcile with the fact that the PC does not differ from the non-target color in this early time range. Hence, while the possibility of an attenuated P1 response cannot be ruled out entirely, it remains a less likely interpretation.

It should also be pointed out that the neural process underlying the prioritized selection of the DC cannot be clarified with the present data. We assume that the early negative polarity modulation represents enhanced neural activity of units coding for the DC, which highlights the representation of the color for subsequent attenuation. It is alternatively possible that the early negativity reflects activity of inhibitory units instantly attenuating the neural representation of the DC. This diminished representation may then be the basis of the subsequent attenuation in the following GFBA time range. In any case, whether the early DC-related modulation is of excitatory or inhibitory nature, it represents a temporal priority signal that highlights the DC for subsequent attenuation.

A notable observation is that while the early DC modulation is generated by source activity in extrastriate visual cortex, it starts (around 75 ms) as transient current maximum in dorsolateral PFC (a region roughly consistent with BA 46/9[22,23]). This is compatible with the dlPFC providing a command for the prioritized distractor representation in the visual cortex and its subsequent rejection. Work in the monkey has shown that FEF controls spatial attention-related activity biases in extrastriate area V4[41–44]. Recently, FEF was shown to command distractor attenuation processes in the visual cortex, as indexed by a monkey homolog of the human Pd[45]—an ERP component reflecting the suppression of salient distractors in visual search[46–54], and which is generated in extrastriate visual cortex[55]. Hence, the initial DC response in PFC could potentially represent an analogous prefrontal control signal for biasing color selectivity in human extrastriate cortex. Notably, the PFC activity observed here arises in dlPFC anterior to FEF (see large 3D view, Fig. 4b), which sets it apart from the frontal control processes just discussed. Nonetheless, lesion studies in humans show that dlPFC as well modulates visual discrimination performance[56]. Activity in monkey dlPFC displays strong distractor suppression that correlates with performance[57]. Finally, monkey dlPFC was shown to encode a

tuned representation of distractor feature values which improves distractor filtering[58]. This could be the basis for a signal rapidly biasing feature representation in extrastriate areas as observed here.

As visible in Fig. 4, the DC modulation in dlPFC (Fig. 4b, yellow dashed) follows the initial feedforward response in V1 (Fig. 4a, red) with seemingly no delay. A higher-resolved comparison of the timing of the respective source waves (Supplementary Figure 2), however, shows that the dlPFC response arises with a small delay relative to the feedforward response in V1. Nonetheless, such small delay cannot be explained by a regular transmission of activity upstream the cortical hierarchy. Response latencies in macaque frontal area FEF (50% response at 75 ms) were also found to arise with only minimal delay relative to V1[59,60], suggesting that cortical bypass connections or direct subcortical projections (e.g., via the mediodorsal nucleus or the pulvinar[61,62]) play a role. Notably, when overtraining monkeys on one color in a color-based search task, color selectivity was seen in FEF already around 70–80 ms after search frame onset[63]. This selectivity follows shortly after the activity onset in V1 and precedes the typical onset of such effects in extrastriate cortex areas, consistent with the latency of the here-observed DC effect in dlPFC. Of course, the locus of source activity in dlPFC seen here rather coincides with the ventral prearcuate (VPA) region than with FEF. However, recent work indicates that area VPA is the source of training-dependent color selectivity in FEF[64], suggesting that color selectivity in VPA would arise at an even shorter latency than in FEF.

Nonetheless, the early onset of the DC-modulation in dlPFC and its fast propagation to extrastriate areas is remarkable, given that a number of cognitive processes (discrimination of the PC, verify the DC, and command suppression of the DC) would have to be put into operation in the short time range of ~10–20 ms. Though the present work cannot clarify the exact nature of the frontal control signal, it is unlikely that such fast distractor selection is the outcome of a time-consuming sequence of decision processes. Instead, we hypothesize that overtraining the two target color sets (used throughout the whole experiment) and the constant spatial layout (target always on the left) enabled subjects to establish a competitive link (opponent coding) between the alternative target colors in dlPFC that rapidly highlights whichever target color appears at the probe location. In fact, such competitive link between the color alternatives presented in the left and right visual field could easily be implemented in the dlPFC. Specifically, the dlPFC displays a bilateral VF representation, with some cells preferring left and other cells preferring right VF targets e.g.,[58]. It has been suggested that this bilateral VF organization may facilitate competitive interactions between items presented in opposite VFs (here: target and probe) via short-range connections, i.e., without the need for callosal transfer. Moreover, target detection by opponent coding has been documented in dlPFC[65], and has been proposed to represent a general efficient mechanism of dynamic coding[66]. Hence, the dlPFC may establish a competitive link between the target color units preferring the left (target) or right VF (probe). This would allow for a fast two-step binary decision process that can operate entirely on the feedforward response of color units in the dlPFC: (1) Determine whether or not both target colors appear on the screen (irrespective of their location), (2) If yes, label the color that is reported by units preferring the probe's visual field (see Supplementary Figure 3 for a schematic illustration). Note, at this early stage of selection, color identity plays no role, the competition process just highlights the target color that happens to appear with the probe even when subjects are not aware which target color is on the target side and which is on the probe side.

Notably, Bartsch et al.[5], using a different version of the UPP did not find such early color biasing (~90 ms after stimulus onset) as observed in the color task here though they also presented distracting colors. In fact, the experimental design varied from the present one, such that early color biasing effects already during the feedforward stage of visual processing would not be expected. Specifically, in Bartsch et al.[5] the target always appeared in combination with an unpredictable color, whereas the target here was either of one or the other target color. Accordingly, a strong preset feedforward sensory-level bias allowing for a fast switch among target alternatives would clash with the target being initially represented as randomly changing color combination during the initial feedforward sweep of processing. Still, future research is needed to determine how the temporal dynamics of early distractor selection for rejection vary with the complexity of visual input and the task at hand (e.g., when attending to conjunctions of different features).

It is worth noting that a similar temporal prioritized selection of distracting items has been reported in visual search[30]. A salient distractor (color singleton) was found to elicit an N2pc-like early (around ~150 ms) component (N1pc) with an onset prior to the same component elicited by the target singleton. Notably, a bigger N1pc response to the distractor singleton was associated with an earlier Pd (distractor positivity) reflecting faster distractor suppression[46,47,52–54,67,68], and was linked to faster target selection. These observations were discussed in terms of prioritized selection for rejection of the distractor singleton, which dovetails with psychophysical work suggesting that the inhibition/deprioritization of distractors in visual search involves the prior attentional selection of those items (attentional 'white bear phenomenon')[69,70] akin to a search and destroy process[69]. The early selection of the DC observed here reflects a conceptually related operation that prioritizes the selection of distracting feature values for rejection, raising the possibility that prioritized selection for rejection is a generic mechanism underlying the dynamic of attentional selection. Further experimental work is necessary to clarify the relation between the N1pc and the early DC modulation.

## Conclusion

Together, our data suggest that trial-by-trial biasing of feature attention 'on the fly' for identifying a target color value among competing task-relevant alternatives almost paradoxically elicits a temporally prioritized neural response to the color alternative that is not the target on a given trial. Notably, a large early modulation to this distracting alternative, followed by a reduced GFBA response to it, is found to facilitate target color identification. This suggests that feature competition is resolved by rapidly indexing the distracting color for neural attenuation prior to target color selection, which preempts interference in this time period. A similar prioritized selection for rejection mechanism has also been documented in visual search for distractors and targets competing as salient singletons in the same feature dimension[30]. Hence, prioritized selection for rejection may represent a more general attention mechanism called upon when competition within a feature dimension needs resolution.

## Methods

**Participants**. Twenty-two volunteers participated in both color and orientation task (mean age 26.0 years, age range 22–33 years, 12 female, all right-handed). All participants had normal or corrected-to-normal visual acuity and reported normal color vision. Participants gave written informed consent prior to the measurement and were paid 6€/h for participation (preparation and measurement lasted 2–3 h). The experimental methods and procedures were approved by the ethics board of the Otto-von-Guericke University

of Magdeburg and conducted according to the research regulations of the Declaration of Helsinki. The number of participants as well as the number of trials per experimental condition (>200) was chosen according to[75], and previous work investigating GFBA components in the EEG/MEG[5,6]. Based on EEG data of a similar experiment[5] (experiment 3), we expected the GFBA effects of interest to be at least of medium-effect size, for which G*Power[81] calculations would suggest twenty subjects to be sufficient for our within-subject repeated measures design (Cohen's $f > 0.25$, power level 0.8, significance level 0.05).

### Experimental design

*Paradigm.* We employed an unattended probe paradigm (UPP), which is a common experimental approach to investigate GFBA in humans and monkeys[9,15,17,19,40]. Participants are asked to covertly attend and discriminate a feature-defined target at a defined spatial location. Meanwhile, an unattended feature probe is presented elsewhere outside the current spatial focus of attention (FOA). Global feature attention will spread constantly across the whole visual field[3,71], including both the location of the target and that of the distant unattended probe. The brain response to the probe is then analyzed as a function of the feature similarity between the probe and the attended target. The response difference between a similar (matching) and a dissimilar (non-matching) feature probe is taken as GFBA effect.

*Stimuli.* The stimuli are illustrated in Fig. 2a. A half-circle in the left visual field (VF) served as target and was presented together with a full circle in the right VF serving as a probe. Target and probe had a circle diameter of 3.1° visual angle and were placed 4.9° lateral and 3.1° below a central fixation cross. On each trial, the target was assigned one of two alternative colors (blockwise either red/green or blue/yellow) and the probe color was randomly chosen from a set of five colors (red, green, yellow, blue, and magenta). The target represented either the left or right half of a circle, i.e., its convexity could either point to the left or right (changing randomly from trial-to-trial). Colors were psychophysically matched prior to the experiment in five independent participants via heterochromatic flicker photometry[72] with an average color luminance of 31 cd/m². The background was dark gray (8.3 cd/m²), the fixation cross white (197 cd/m²).

*Procedure.* Participants were covertly attending to the target half-circle in the left VF, while their fixation remained on the central cross. The participants were either asked to report the color of the target (red or blue: index finger, green or yellow: middle finger), or to report its orientation (convexity left: index finger, convexity right: middle finger). Task and target colors were designated at the beginning of each trial block. Specifically, the initial instruction screen depicted all possible four target half-circles for a given block (e.g., red and green, convexity left and right) with the respective correct responses. To avoid any confusion with or priming of the non-target colors we did not explicitly inform participants about the identity of the non-target colors on a given block. Every subject performed six color and six orientation blocks (each block lasting about 5 min) in one of four possible pseudorandomized orders, such that the color of the targets (red/green or blue/yellow) was never repeated on subsequent blocks and the task (color or orientation) changed every second block. Target and probe were simultaneously presented for 300 ms, followed by an interstimulus interval with only the fixation cross being present (randomly varying between 1000 and 1200 ms, rectangular distribution). Each subject performed 180 trials per experimental block, yielding at least 216 trials for the individual trial types (averaged across different colors, see below).

*Task types.* In different experimental blocks, participants discriminated either the color (color task, attentional two-color template), or the orientation of the target (orientation task, color irrelevant). For both tasks, stimulus geometry and stimulus timing were kept identical, such that only the attention condition but no physical stimulation properties differed between task blocks. The purpose of the color task was to investigate the dynamics of biasing color selection for two colors 'on the fly' the moment target color A or B appears. The orientation task served as a control condition with color being irrelevant. That way, we could separate the effects of attentional color biasing from the effects of color selection that would also appear when color is irrelevant, e.g., caused by target discrimination itself, color priming[24,27], or even low-level stimulus properties. In fact, any discrimination performed on the target will most likely entail some processing of its color as would be revealed by the orientation task.

*Trial types and GFBA derivation.* The probe's color could match the color of the currently presented target half-circle (present target color, PC), match the distracting alternative target color (DC), or neither of them (non-target color). On a given trial block, there were always three non-target colors, i.e., the two colors that would be a target on the other blocks, and magenta that was added to introduce a greater variety of probe colors. However, since magenta was never used as a target, it is the only color where color-specific effects could not be controlled for (no attended vs. not attended comparison possible), such that magenta probes were excluded from the main analyses. The brain response to magenta compared to that of the other non-target colors is shown and discussed in the Supplementary Information (Supplementary Note 1, Supplementary Figure 4). Figure 2b illustrates examples of the different trial types for a red probe. The effects of global feature-based attention (GFBA) were assessed by measuring the brain response to the color probe in the unattended hemifield (unattended probe paradigm, UPP). The brain response elicited by probes containing attended colors, i.e., the PC or DC, was then compared with responses to probes drawn in unattended colors not relevant in the current experimental block (non-target colors, serving as baseline condition), see also previous work[5,6,18,19]. Note that any comparison between attended (DC, PC) and unattended (non-target) probe colors involves identical attention conditions at the target side (target always drawn in PC). Hence, attention-related response differences are expected to largely cancel contralateral to the target (see Supplementary Figure 5 for ERPs recorded contralateral to the target). To increase the signal-to-noise ratio and to also avoid color-specific confounds, brain responses were averaged across different probe colors (red, green, blue, and yellow) within trial types. As visible in Fig. 2b, the response to the same physical color probe could be compared under different attention conditions. However, the simplicity of the experimental design with a mono-colored target entailed limitations of color balance between trial types. Specifically, only when the probe matched the current target, the same color was present in both visual fields. Nevertheless, as shown in previous experiments, the feature bias for this trial type cannot be attributed to low-level stimulus properties. Without attention to color, the mere presence of the same color on both sides of the visual field does not itself give rise to GFBA or GFBA-like modulations (experiment 4 of Bartsch et al.[5], experiment 1 of Boehler et al.[73], separate object condition).

### Data acquisition

Participants were equipped with a 32-electrode cap and seated in a dimmed, electrically, and magnetically shielded recording booth (Vacuumschmelze, Hanau, Germany) below a MEG dewar in front of a partly transparent screen

(COVILEX GmbH, Magdeburg, Germany) at a viewing distance of 1.0 m. Stimuli were backprojected onto the screen using an LCD projector (DLA-G150CLE, JVC, Yokohama, Japan) placed outside the booth. Stimuli were delivered using Presentation® software (Neurobehavioral Systems Inc., Albany, CA). Participants gave responses with their index and middle finger of the right hand using a LUMItouch response system (Photon Control Inc., Burnaby, DC, Canada).

*EEG/MEG recording.* The electroencephalogram (EEG) was continuously recorded using a 32-electrode cap with mounted sintered Ag/AgCl electrodes (Easycap, Herrsching, Germany) and a Synamps amplifier system (NeuroScan, El Paso, TX). Electrode positions were chosen according to the international extended 10–20 system[74]. Contact between electrodes and head surface was established using the abrasive electrolyte gel Abralyt light (Easycap, Herrsching, Germany), impedances were kept below 5 kΩ at all electrode positions. An electrode at the right mastoid served as online reference during recording, data were then offline re-referenced to the weighted mean of the electric activity of the electrodes at the left and right mastoid according to[75]. To track eye movements, electrodes were placed at the outer canthi of both eyes (bipolar derivation) and an electrode placed below the right eye (unipolar derivation) recording both the horizontal and vertical electrooculogram (EOG). The magnetoencephalogram (MEG) was simultaneously recorded with a 248-sensor BTI Magnes 3600 whole-head magnetometer system (4D Neuroimaging, San Diego, CA, USA). Environmental noise was canceled online using built-in reference coils[76]. To coregister individual head positions with the MEG sensor array, anatomical landmarks (left and right preauricular point, nasion) and five localizer coils placed on the EEG cap (near inion, vertex, nasion, left and right preauricular point) were digitized using a Polhemus 3Space Fastrak System (Colchester, VT, USA). Since head positions varied among participants, individual sensor data were repositioned offline with reference to a canonical head-sensor configuration (average of 1500 individual coregistrations). To this end, each participant's leadfield was calculated with Curry 7 Neuroimaging Suite (Compumedics Neuroscan, Compumedics USA, Ltd., Charlotte, NC, USA) using sensor data and the MNI brain (Montreal Neurological Institute brain, ICBM-152 template). By (pseudo-)inverting the leadfield (minimum norm least-squares approach), individual MEG sensor data were then transformed into MNI source space and afterward backprojected into the sensor space of the canonical head-sensor configuration using the respective reference leadfield.

EEG, MEG, and EOG data were band-pass filtered (DC-50Hz) and digitized at a sampling frequency of 254.31 Hz. EEG data were used to investigate event-related potential (ERP) components, MEG data served to estimate the underlying current sources (as described in Data analysis).

*Control of eye movements.* To prevent participants from breaking fixation or directly looking at the stimuli and/or position of the upcoming target, we took the following precautions: at the beginning of the experiment, participants were instructed to keep their eyes all the time on the central fixation cross (except for blinking pauses or breaks between blocks). They then performed a couple of test trials until they felt comfortable solving the task without directly looking at the target stimulus. During the measurement, the experimenter closely monitored eye position using video surveillance and the EOG signal. Trials with eye movements that took place in the analyzed epochs (200 ms before to 700 ms after stimulus onset) where excluded based on the horizontal and vertical EOG (see below EEG/MEG—epoching and artifact rejection). None of the

participants showed excessive eye movements or repeatedly failed to maintain fixation.

### Data analysis

*Behavioral data.* Response time and response accuracy were computed using MATLAB routines (MathWorks Inc., Natick, MA, USA), respective temporal onsets and identities of stimuli and given responses were derived from the logfiles produced by Presentation® software. Trials with anticipatory (<200 ms) and delayed (>1300 ms) responses were excluded, and only correct responses were used for response time measures. For statistical validation, the data were analyzed with the software package SPSS (SPSS Inc., Chicago, IL, USA) by computing repeated measures ANOVAs (rANOVAs). Significant main or interaction effects were further evaluated using subsequent paired Student's *t*-tests. An alpha of 0.05 served as significance level, Greenhouse–Geisser correction was applied to correct for nonsphericity.

*EEG/MEG—epoching and artifact rejection.* The continuous EEG and MEG data were epoched offline from 200 ms before stimulus onset to 700 ms after stimulus onset. After excluding anticipatory (<200 ms), delayed (>1300 ms), and incorrect responses, epochs were subjected to an artifact rejection. All epochs exceeding specific peak-to-peak amplitude measure thresholds were removed until the data were devoid of major artifacts including eye blinks, eye movements, and physiological noise like muscle tension. To this end, data were visually inspected and thresholds individually determined for every subject ranging from 70 to 115 μV (mean 97.5μV) for EEG and 2–3.5 pT (mean 2.9 pT) for MEG. This led to on average 5.4% (EEG) and 6.6% (MEG) rejected trials.

*Event-related potentials (ERPs).* After artifact rejection, the remaining EEG/MEG epochs (including only correct responses) were averaged locked to stimulus onset for the individual trial types within participants. Given the low number of incorrect responses (>92% correct responses for all conditions), and the resulting insufficient signal-to-noise ratio, ERPs for incorrect trials were not analyzed. Shown electrode sites and time ranges were chosen in accordance to previous work[5,6,18,19]. Specifically, GFBA effects are expected to appear between 150 and 300 ms contralateral to the unattended probe at parieto-occipital sensor sites. More specifically, due to the contralateral retinotopic organization of the visual cortex, a probe in the right VF will elicit maximal GFBA activity in the left ventral occipital cortex, that can typically be measured best at electrodes at parieto-occipital recording sites placed over the left hemisphere (i.e., PO3, PO7, and PO9). A visual inspection of the data confirmed prominent GFBA modulations for color in the expected N1/N2 time range. As can be seen in the topographical EEG maps (Fig. 3/7 b, c), effect maxima are fairly comparable to previous work with electrodes PO3 and PO7 appearing closest to topographic field maxima across all conditions. Hence, the signal was averaged across the respective electrode sites for the reported ERP waveforms. ERP waveforms are plotted from −150 to 300 ms with the 150-ms prestimulus period serving as a baseline for all analyses. ERP waveforms and topographical field distribution maps were plotted using the Event-Related Potential Software System ERPSS (Event-Related Potential Laboratory, University of California San Diego, La Jolla, CA, USA). A smoothing Gaussian filter (low-pass, half-amplitude cutoff frequency of 23 Hz) was applied for visualization purposes only, statistical testing was performed on unfiltered data. All waveforms and topographical maps display 'grand average' data (i.e., data averaged across all twenty-two participants).

*Statistical validation of amplitude differences.* To retrieve the time course of GFBA modulations and define time windows of significant differences between the conditions, a time-sample-by-time-sample sliding 2×3 rANOVA was performed with the factors TASK (orientation, color) and COLOR (probe color matches PC, DC, or non-target). The data were tested in the time range of 0–300 ms after stimulus onset, the width of the sliding window was 11.8 ms (i.e., three time samples). To correct for multiple comparisons, we followed the logic of Wagner et al.[77] taking into account the original sampling frequency ($f_s$, here 254.31 Hz) and the applied low-pass filter ($f_c$, here 50 Hz). The corrected alpha level was $1-(1-0.05)^{2f_c/f_s} \approx 0.02$. The first out of five or more successive sample points with a *p*-value below 0.02 was considered as effect onset. All subsequent statistical comparisons were performed within time ranges of significant main or interaction effects. For further explorative analyses, additional sliding t-tests between single conditions were performed outside the predefined time windows.

*Current source localization.* Current source activity underlying modulations of the ERP components was computed based on the simultaneously recorded MEG data, which provide spatial resolution superior to the ERP data. Current sources were estimated using a distributed source model computed on the grand average data using the minimum norm least-squares approach as implemented in the Curry 8 Neuroimaging Suite (Compumedics Neuroscan, Compumedics USA, Ltd., Charlotte, NC, USA)[78]. The standard MNI-152 brain included in Curry 8 (Standard Cortex) was used as source compartment, which provides a gray/white matter border triangularization, resulting in 20006 fixed current dipole locations. To assess the localization of current sources relative to the retinotopic areas of the visual cortex, we defined regions of interest (ROIs) around source maxima above an arbitrary threshold to minimize overlap among successively activated areas. The ROIs were then compared to the location of retinotopic areas (FEF, V3v/d, and hV4) taken from probabilistic maps of visual topography in the human cortex available at https://scholar.princeton.edu/napl/resources [79,80]. The retinotopic areas were rendered onto a 1-mm surface segmentation of the MNI152 brain using FreeSurfer (version 5.1.) and FSL (http://www.fmrib.ox.ac.uk/fsl/).

*Median response time-split analysis (RT split).* Data were reanalyzed to separate between slow and fast responses for specific trial types. To prevent any contamination by other influences like a particular color or response button (e.g., responses might be faster when the target is blue compared to green, or when answering with the index finger compared to the middle finger), the split was performed for every single stimulus display (each stimulus was shown 27 times, e.g., red target facing to the left with probe being green). Afterward, all trials tagged as "slower" or "faster" than the median RT for that specific stimulus were averaged together with the respective "slow" or "fast" trials of the other stimuli contained in a specific trial type.

*Target repetition analysis.* Data were analyzed as a function of whether the target color was identical to that of the previous trial (target-repetition trials), or changed (target-switched trials). Since target color was randomly drawn from the two possible target colors on every trial, the analysis yielded equally sized bins for the conditions (same probability for each trial to be a repeat or switch trial). Analogous to the RT split analysis, trial split was performed for every single-stimulus display before averaging together the trials for the individual trial types (e.g., PC/DC).

*Statistics and reproducibility.* Sample sizes and statistical tests are indicated in the text and figure legends. In general, the number of participants and trials was chosen according to previous work[5,6]. The sample size was also in accordance with calculations of G*Power[81]. The participants performed two tasks (orientation/color) with three experimental conditions (TC, DC, non-target) each. Behavioral effects as well as time ranges of significant neural responses were determined using 2×3 repeated measures ANOVAs (within-subject design), and subsequent paired t-tests. When sliding-window tests (ANOVA or t-test) were conducted on EEG data, we corrected for multiple comparisons following the logic of Wagner et al.[77]. Current source estimates on MEG data were calculated using a minimum norm least-squares approach[78] and the standard MNI-152 brain. More detailed information on statistics is provided in the corresponding methods sections above. GFBA modulations for color as assessed by the unattended probe paradigm here have been replicated in several previous experiments[5,18,19].

**Reporting summary**. Further information on research design is available in the Nature Research Reporting Summary linked to this article.

## Data availability

The datasets generated during and/or analyzed during this study are available from the corresponding author on reasonable request. Source data underlying the graphs and charts presented in the main figures are made freely accessible at https://osf.io/chbnd/.

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

## Acknowledgements
We thank Steffi Bachmann and Laura Hermann for assistance with the data acquisition as well as Hendrik Strumpf for support in data analysis. This work was supported by the Deutsche Forschungsgemeinschaft (Grant SFB779/TPA1).

## Author contributions
M.V.B. and J.M.H. planned the experiments. M.V.B. performed the research. M.A.S. provided scientific support. M.V.B. wrote the paper. J.M.H. and C.M. edited and reviewed the paper.

## Funding

## Competing interests
The authors declare no competing interests.
