## [Peer Review File · Communications Biology]

Reviewers' Comments:

Reviewer #1:

Remarks to the Author:

Bartsch et al. use a modified version of the unattended probe task in combination with EEG/MEG to investigate the cortical dynamics involved in adjusting color selectivity between unpredictable targets. Participants had to covertly attend and discriminate the color of a target located in the left visual field, while a distractor was placed in the opposite position. The target color was selected at random in two possible colors and the distractor could be one of five colors (the 2 target colors and 3 non-target colors). This led to 3 different type of trials. Trials where the distractor had the same color as the target (PC trials), trials where the distractor had one of the two target colors but was a different color than the current target (DC trials), and control trials where the distractor was one of the non-target colors.

The authors report that PC trials evoked a longer and more negative ERP response in parieto-occipital electrodes sites contralateral to the distractor. In DC trials this global feature based attention (GFBA) response was smaller in amplitude and preceded by an earlier negativity enhancement generated by source activity in dIPFC. The amplitude of this DC early negativity was inversely related to the participant's response reaction time. The authors suggest that the DC early negativity helps the suppression of the late GFBA response and facilitates the rejection of the distractor.

The experiment is well designed, and the findings are clear. The paper is also well written, and the flow of ideas is easy to follow. The authors did a good job situating their work in the context of previous research. I only have a couple general issues and a handful of specific/minor issues for the authors to address.

GENERAL ISSUES:

- One of the main findings of the study is that the early DC negativity is generated by source activity in dIPFC. The authors interpreted these results as evidence of a top down modulation to bias color selectivity in visual cortex. Ultimately, this facilitates an early rejection of the distractor. This is an interesting finding, however; the authors only show the results for the DC trials and not the trials where the target and the distractor had the same color (PC trials). In figure 3B (bottom panel) the authors show a smaller early negativity during PC trials. If this early source activity in dIPFC is modulating the early negativity enhancement in DC trials, we should observe differences in the amplitude of this source activity between DC and PC trials.

- For the median response time split analysis, categorizing a continuous variable by splitting them at their median is problematic because of the risk to increase Type II errors. Keep the reaction time as a continuous variable and then use a linear regression would be a more appropriate method to analyze the effects of the GFBA amplitude in task performance.

MINOR ISSUES:

- Were the participants informed about the target and non-target colors before the start of each block?

- Even when participants show a high behavioral performance (over 90% correct response for all trial types), it would be interesting to see if there are any differences in the ERP waveforms between correct and incorrect response trials. In particular, it would be interesting to see if the early negativity during DC trials is also related to this behavioral performance.

- The magenta distractor was the only true task-irrelevant color across all blocks. All the other colors were used as target colors in different blocks of trials. However, the authors excluded all trials where the magenta distractor was present and argue that during these trials color-specific effects could not be controlled. If the magenta probe was presented with the same probability than the other non-target colors, the magenta trials would be the best trials to use as controls. Do the

ERP waveforms between the magenta trials and the other non-target colors trials differ?

- Participants covertly attended to a peripheral target while fixating a central cross. However, the authors did not use an eye tracker to monitor fixation. EOG was used to detect eye movement related artifacts in EEG/MEG epoch data. The authors should address if any trials were excluded from the main analyses because of larger EOG signal before the epoch window, which might indicate breaking fixation before the attention period.

Reviewer #2:

Remarks to the Author:

Bartsch and colleagues investigated the Global Feature-Based Attention (GFBA) effect whereby attending to a target stimulus of a particular color in one visual field results in enhanced processing of probes of that color in the opposite (unattended) visual field. This effect was manifested in previous studies by this group as an enhanced negativity at 150-250 ms in the event-related potential (ERP) recorded over the occipital scalp contralateral to the probe that matched the attended target color. The current study was designed to examine the processing of a probe having an alternative distractor color in the unattended visual field. Surprisingly, the distractor probe elicited a very early negative enhancement onsetting before 70 ms in relation to the ERP elicited by a "neutral" probe of a non-distractor color. This early negativity was enlarged on trials having faster RTs for discriminating the target color and on trials where the target color was repeated, and it was absent in a task where the same stimuli were discriminated for their spatial position rather than color. The authors interpret this early negativity, which was source localized initially to the dorsal lateral prefrontal cortex (DLPFC), as indicative of an inhibitory process that suppresses the competing distractor color information. It was suggested that the DLPFC initiates a command for an early "selection for rejection" process.

This study is well designed and methodologically rigorous. If the authors' interpretation of the early negativity is correct it would represent a remarkable new early inhibitory mechanism. I have some questions, however, about the timing of this early ERP modulation. Figure 4A shows that the initial feed-forward activation localized to area V1 begins at about 65 ms and is still small at 75 ms, whereas Figure 4B shows that the activation of the DLPFC, which also begins at about 65 ms, has nearly reached its maximum by 75 ms. So it seems that the frontal activation is nearly instantaneous with (or even prior to) the initial activation in primary visual cortex. How can this be? It would seem that initiation of a distractor suppression process would involve a sequence of events: (1) discrimination of the target color, (2) linking this target color with its alternative distractor color, a linkage that depends on memory and possibly inter-hemispheric information transfer and (3) initiation of a suppression of the distractor. How can all this processing possibly take place in just a few ms? The transmission time from V1 to the DLPFC also needs to be taken into account. The authors need to explain how this selection for rejection could work within the time frame of the source waveforms shown in Fig 4.

Looking at the literature, it seems that overt attention to a specific color generally modulates the ERP starting at around 100 ms (e.g., Anllo-Vento et al. Human Brain Mapping 1998; Schoenfeld et al. Cerebral Cortex 2007). The earliest modulation I could find was found by Zhang & Luck (2009), who found that an attended-color probe elicited an enhanced positivity onsetting at 80 ms. In all these studies a particular color was attended top-down, so that color-specific feed-forward pathways could be primed before stimulus delivery, and therefore a very early modulation of the feed-forward sweep was possible. In the present study, however, it is not possible to attend to a specific color but rather to discriminate between two alternative colors. How is it possible for such discrimination plus the additional steps listed above to be achieved faster than simple top-down feed-forward biasing? The authors need to explain further how such a proposed distractor suppression mechanism could possibly work within the observed time frame.

It would be of interest to examine the ERPs recorded over the hemisphere contralateral to the targets on exactly the same trials where the early negativity was revealed contralateral to the distractors. Surely the authors have looked at this. I assume they would predict no early negativity in the appropriate difference waves. If there were such an early modulation it may require some re-thinking of the suppression hypothesis.

A few minor wording improvements:

--Line 344: change "would require" to "would be required"
--Line 362: change "enforces" to "reinforces"
--Line 374 "...which was much smaller..."
--Line 759: "...relative to the..."

Reviewer #3:

Remarks to the Author:

The current study assessed whether feature-based enhancement of target colors and suppression of distractor colors. Participants performed a task where they classified the orientation of a circle and tried to ignore a distractor in the opposing hemifield. The distractor could be the presented target color, a distractor color, or a neutral color. EEG and MEG were measured in response to these stimuli. Three experiments seemed to indicate that the representation distractor was initially enhanced before it was suppressed.

The results are interesting and address an important question. However, I have some reservations about the interpretation of the ERP waveform. Namely, one could argue that the early negativity toward the distracting color as indicates attentional suppression, not attentional enhancement of the distractor.

Major Comments:

1. The authors argue for a "selection for rejection" based upon an early negative enhancement in the ERP waveform, which they propose reflects attentional enhancement of the distractor probe. However, many previous studies have used P1 enhancement effects as an indicator of feature-based attention (for reviews, see Hillyard et al., 1998; Mangun, 1995). The basic pattern of results in these studies is that the P1 wave is larger (i.e., more positive) for attended items than unattended items (Heinze et al., 1994). As shown in Figure 3, the distractor colored probes (DC) produce a smaller P1 than the non-target baseline. In my view, this relative negativity seems more consistent with the interpretation that attentional allocation was reduced for the distractor color (DC) probes compared to baseline. In other words, could argue that the distractor color was initially suppressed, not attended as the authors have claimed. At a minimum, some clarification of the current results in the context of P1 enhancement is needed. But ultimately a strong reinterpretation of the data seems warranted.

2. The above P1 reinterpretation could also be applied to the other results. For example, in Figure 5, this new interpretation would suggest that successful early suppression of the distractor color (DC) probe resulted in a faster the RTs to detect the target stimulus (Figure 5). In other words, if the negativity in the P1 range indicates suppression (rather than attentional allocation), then the fast RT trials would be the trials with successful early suppression of the distractor color.

3. Again, if you assume that relative negativity (compared to baseline) in the P1 range reflects suppression (i.e., less P1 enhancement), Figure 6 would suggest that repeating the target color causes the DC to be suppressed not enhanced.

Minor Comments

Figure 5: For the early vs. late analysis, might it be useful to show the actual difference waveforms for fast vs. slow RT (instead of just the bar plots). If they are extremely noisy, the authors might consider including them as a supplement.

p. 14 When introducing the idea of intertrial color priming, it seems important to cite the seminal studies of Maljkovic and Nakayama (1994, 1996).

p. 15 Perhaps the "selection for rejection" mechanism being proposed is also similar the "search and destroy" models proposed by Egeth and colleagues.

Reviewers' comments:

Reviewer #1 (Remarks to the Author):

Bartsch et al. use a modified version of the unattended probe task in combination with EEG/MEG to investigate the cortical dynamics involved in adjusting color selectivity between unpredictable targets. Participants had to covertly attend and discriminate the color of a target located in the left visual field, while a distractor was placed in the opposite position. The target color was selected at random in two possible colors and the distractor could be one of five colors (the 2 target colors and 3 non-target colors). This led to 3 different type of trials. Trials where the distractor had the same color as the target (PC trials), trials where the distractor had one of the two target colors but was a different color than the current target (DC trials), and control trials where the distractor was one of the non-target colors.

The authors report that PC trials evoked a longer and more negative ERP response in parieto-occipital electrodes sites contralateral to the distractor. In DC trials this global feature based attention (GFBA) response was smaller in amplitude and preceded by an earlier negativity enhancement generated by source activity in dlPFC. The amplitude of this DC early negativity was inversely related to the participant's response reaction time. The authors suggest that the DC early negativity helps the suppression of the late GFBA response and facilitates the rejection of the distractor.

The experiment is well designed, and the findings are clear. The paper is also well written, and the flow of ideas is easy to follow. The authors did a good job situating their work in the context of previous research. I only have a couple general issues and a handful of specific/minor issues for the authors to address.

GENERAL ISSUES:

- One of the main findings of the study is that the early DC negativity is generated by source activity in dlPFC. The authors interpreted these results as evidence of a top down modulation to bias color selectivity in visual cortex. Ultimately, this facilitates an early rejection of the distractor. This is an interesting finding, however; the authors only show the results for the DC trials and not the trials where the target and the distractor had the same color (PC trials). In figure 3B (bottom panel) the authors show a smaller early negativity during PC trials. If this early source activity in dlPFC is modulating the early negativity enhancement in DC trials, we should observe differences in the amplitude of this source activity between DC and PC trials.

Response: We completely agree with the reviewer that it is important to show source activity in dlPFC also for PC trials, especially since the ERP waveform of the PC trials shows some smaller early modulation. We accordingly added the time course of source activity elicited by PC probes in the ROIs of the DC response in Figure 4B. As can be seen, except for an initial weak modulation present in all ROIs, there is no significant source activity in dlPFC. This weak modulation does not differ among cortical regions, indicating that it essentially reflects noise fluctuation. As the reviewer predicted, we found substantial differences in dlPFC source activity between DC and PC trials, verifying that the dlPFC activity exclusively appears in response to the DC. The results of the source analysis in PC trials are now discussed in the results section.

Figure 4. Time course of cortical current source activity for the color task. (A) Shown is the propagation of stimulus-elicited activity in visual cortex after stimulus onset (current source activity of the average across target (PC), non-target and distractor (DC) color probes). The waveforms show time courses of source strength at selected ROIs (red: primary visual cortex, orange: lateral occipital (LO) cortex, green: IT cortex) (see Methods). Small 3D views show current source density maps at selected time points illustrating the initial feedforward-sweep of processing. **(B)** Time course of the source activity underlying the very early distractor selection (DC minus non-target, color-dashed shown between 40-150ms) at selected ROIs depicted in middle-sized and large 3D view (yellow: dorsolateral prefrontal cortex, green: IT cortex, blue: V4, red: V3). For target-colored probes, the time course of source activity at the respective ROIs does not show comparable modulations (PC minus non-target, color-solid). Small 3D views show source density maps at selected time points illustrating the prefrontal-to-ventral extrastriate propagation of the DC biasing. As can be seen, the very early selection bias for the distracting color appears already during the initial feedforward-sweep of information processing (compare time range highlighted by grey horizontal rectangles in A and B).

Figure changes: The source waveforms of the PC trials are added to Figure 4B.

Text changes: l.272-275: "Importantly, when analyzing the source activity in the same ROIs for the PC, this early sequence of source activity in prefrontal-to-ventral extrastriate cortex was absent (Figure 4B, color-solid). Specifically, the PC source activity showed only a weak early peak that was simultaneously present in all ROIs, presumably reflecting noise fluctuations."

- For the median response time split analysis, categorizing a continuous variable by splitting them at their median is problematic because of the risk to increase Type II errors. Keep the reaction time as a continuous variable and then use a linear regression would be a more appropriate method to analyze the effects of the GFBA amplitude in task performance.

Response: The reviewer is right that dichotomizing continuous predictor variables by splitting them at their median can in fact decrease the statistical power. Especially if you expect a linear relationship between response time and ERP amplitude, a linear regression account might seem more suitable compared to a median split. However, single-trial ERP analyses are fundamentally limited by their signal-to-noise ratio. This is why we averaged across many trials (50-100) to get a reliable estimate of the amplitude effect (Luck 2005). Specifically, a single trial will not show a clear modulation in the expected time range at all as it is dominated by a spontaneous fluctuation of activity, typically magnitudes larger than the signal. This is an issue hard to come by even with advanced signal denoising methods. Furthermore, the signal under consideration here is derived as ERP difference (PC or DC minus non-target color). Such difference cannot be derived from single trials. Finally, a single-trial based RT regression analysis would face another issue. The response time on a given trial is also influenced by unspecific factors that are under control when averaging trials according to experimental conditions, but that would confound the analysis when considering single trials. For example, in previous work using the UPP (Bartsch et al. 2015, supplementary materials) we found that right-facing targets are responded to faster than left-facing targets, such that a slow response to the former may be as fast as a fast response to the latter. To eliminate the influence of such factors in our median-split analysis, we performed the split for each target color and orientation separately. One way to control for such unspecific RT effects in a single-trial based RT regression analysis would be to equal the RT range (z-transform) of the different target colors and orientations. Of course, this introduces its own set of issues. Nonetheless, we have performed a single trial regression analysis as suggested by the reviewer based on z-transformed RT-data of DC-trials (single-trial ERPs were wavelet-denoised, Quiroga & Garcia, 2003 Clin Neurophysiol). The Figure below shows the result of an example subject. As expected, while the RT-amplitude link is clearly present (the median split is highlighted in color: red-slow, green-fast), the amplitude fluctuation of the denoised response between 50-100ms is, however, enormous relative to the experimental effect, placing the correlation in a range of ~4% of the total variance.

Single trial regression analysis shown for one example subject. Shown is a scatter plot of the amplitude of DC trials (mean amplitude in the early time range, 73-96ms) and z-transformed response times. Single trials were wavelet-denoised (Quiroga & Garcia, 2003 Clin Neurophysiol). Green dots indicate "fast" and red dots "slow" trials as sorted by a median split, black dots depict the respective mean of fast and slow trials. The regression line indicates that a more negative amplitude is associated with faster response times. Due to the huge amplitude fluctuations on single trials, this relationship fails to reach significance.

We are, however, aware that a median split is a robust and conservative but rather coarse measure. To provide a more fine-grained analysis, we now added a quartile-split analysis as Supplementary Figure S1 and refer to it in the results section. As can be seen, the early amplitude is in fact gradually reduced in amplitude with increasing response time.

Supplementary Figure S1. Quartile response time (RT) split for DC trials. The quartile RT split was performed analogous to the median RT split as detailed in the Methods section of the manuscript. **(A)** ERP waveforms for DC trials (probe matches distracting target color alternative) split into four quartiles (very fast, fast, slow, and very slow RTs). As can be seen, in the time range of the distractor negativity (black rectangle), the response shows an increasing negativity with decreasing RT. **(B)** Mean GFBA amplitudes (DC minus non-target differences) of the quartile responses in the time range marked by the black rectangle in A (57-96ms). The early modulation effect is gradually reduced in amplitude with increasing response time, with a large amplitude effect appearing in very fast (1st quartile, dark blue) and a smaller effect in fast trials (2nd quartile, light blue). The early DC modulation is minimal in slow (3rd quartile, orange) and absent in very slow trials (4th quartile), indicating that it is primarily driven by fast target discriminations.

Figure changes: The RT-quartile split is added as Supplementary Figure S1.

Text changes: l.319-325: “A further RT-quartile split analysis of the data shown in Supplementary Figure S1 qualifies this observation. The early modulation effect is gradually reduced in amplitude with increasing response time, with a large amplitude effect appearing in very fast (1st quartile) and a smaller effect in fast trials (2nd quartile). The early DC modulation is minimal in slow (3rd quartile) and absent in very slow trials (4th quartile), indicating that it is primarily driven by fast target discriminations.”

MINOR ISSUES:

- Were the participants informed about the target and non-target colors before the start of each block?

Response: The participants were informed about the target colors before the start of each block. Specifically, the instruction screen at the beginning of the block depicted the possible target half circles (e.g., red and green) with the respective correct responses. We did not explicitly inform participants about the identity of the non-target colors on a given block. In the orientation task, the instruction screen showed the same colored half circles but now with the instruction to respond to their orientation.

Text changes: We added this information to the methods section (l. 698-702). It now reads: “Task and target colors were designated at the beginning of each trial block. Specifically, the initial instruction screen depicted all possible four target half circles for a given block (e.g., red and green, convexity left and right) with the respective correct responses. To avoid any confusion with or priming of the non-target colors we did not explicitly inform participants about the identity of the non-target colors on a given block.”

- Even when participants show a high behavioral performance (over 90% correct response for all trial types), it would be interesting to see if there are any differences in the ERP waveforms between correct and incorrect response trials. In particular, it would be interesting to see if the early negativity during DC trials is also related to this behavioral performance.

Response: We agree with the reviewer that it would indeed be interesting to compare waveforms between correct and incorrect trials, especially with respect to the early biasing. Unfortunately, the small number of trials prevents a reliable amplitude estimate for incorrect trials. That is, not taking into account artifact rejection, participants had on average only about 16 incorrect DC trials, which does not suffice for a reliable ERP amplitude estimate, especially since with such small number, the amplitude will be strongly influenced by noise (Luck 2005). For the other trial types, there are even fewer trials. Nonetheless, we thank the reviewer for pointing this out, and we think that with employing a more challenging task, it will be possible in the future to look at the influence of task-difficulty on the early biasing with a reasonably high number of incorrect trials.

Text changes: We added a sentence to the Methods section (l.831-833). “Given the low number of incorrect responses (> 92% correct responses for all conditions), and resulting insufficient signal-to-noise ratio, ERPs for incorrect trials were not analyzed.”

- The magenta distractor was the only true task-irrelevant color across all blocks. All the other colors were used as target colors in different blocks of trials. However, the authors excluded all trials where the magenta distractor was present and argue that during these trials color-specific effects could not be controlled. If the magenta probe was presented with the same probability than the other non-target colors, the magenta trials would be the best trials to use as controls. Do the ERP waveforms between the magenta trials and the other non-target colors trials differ?

Response: The reviewer brings up an interesting point. Indeed, the lack of control for color-based effects was the main reason for excluding magenta trials. That is, with magenta never being the target, there can be no “attend” minus “not attend” difference for magenta probes, that would eliminate low level effects of this color. In particular, any effect reported for magenta could end up being color-specific (i.e., just occurring for magenta and not representing a general effect of a “neutral color”). Nevertheless, we agree that it can be informative to analyze the brain response to magenta probes, since it is a color that never gained task-relevance. We analyzed the respective brain response to magenta probes and compared it to those of the other non-target colors (now shown as Supplementary Figure S4).

As can be seen, the magenta response (magenta line) is quite similar to that of the non-target colors (black dashed) and does not significantly differ in the time range of the early biasing effect around 90 ms after stimulus onset. However, as best depicted by difference waveforms

(panel B), irrespective of the task, the non-target colors show some negative enhancement compared to magenta around 150ms (sliding-window t-test reveals significant differences between about 136-170ms). In other words, in that time range, the brain response to magenta is “below” of that to non-target colors. One possible interpretation is that by the virtue of being target colors on other experimental blocks, the non-target colors gained some overall bias throughout the experimental session resulting in a small GFBA effect (negativity around 150 ms) relative to magenta. Alternatively, magenta could be actively suppressed relative to the non-target colors. The distracting potential of magenta might stem from two different sources: first, magenta is the color half-way between red and blue, such that it might have interfered with internal representations of those colors. Second, though magenta was shown with the same frequency on the probe side, it never appeared on the target side, rendering it overall less-likely to appear on a given trial. Hence, magenta will potentially have been perceived as a rare “popout” color. We now show and discuss the response to magenta probes as part of the Supplementary materials.

Supplementary Figure S4. ERP results for non-target magenta-colored probes. (A) Shown is the ERP elicited by the probe at PO3/PO7 (signal averaged) for PC, DC, and non-target probes (replotted from Figure 3 and 7) together with the ERP elicited by non-target magenta probes when participants were to discriminate the color (upper row) or orientation (lower row) of the target. Rectangles highlight time ranges of significant brain response variations as defined by a 2x3 rANOVA (see Methods, main text). As can be seen, the brain response to magenta probes (magenta line) is quite similar to that of the other non-target colors (black dashed). **(B)** Difference waveforms of non-target minus magenta for the color (upper row) and orientation task (lower row) reveal a greater brain response (negativity) for non-target colors peaking around 150ms. Irrespective of the task, sliding-window t-tests (sample-by-sample, 11.8ms window between 0-300ms) confirm a significant difference of non-target and magenta between about 136-170ms after stimulus onset (magenta horizontal bars) but find no significant differences in the time ranges of our main analysis (rectangles). The respective topographical maps on the right display the field distribution at representative time points (early, middle, late). The positions of electrodes used for the analyses are highlighted (black ellipses). The magenta-colored non-target probes do not qualitatively differ from the other non-target colors in the early and late modulation time ranges of our main analysis. However, around 150ms, the non-target colors show some negative enhancement relative to magenta, potentially due to being task-relevant on other experimental blocks.

Importantly, except for adding the small modulation effect around 150ms, none of the reported effects would qualitatively change when using magenta as a reference.

Of note, we recently ran an experiment where we systematically looked for differences between entirely neutral versus previously relevant non-target colors (data yet unpublished). In this experiment, the neutral color changed between subjects, was not similar to any of the target colors, and could also appear as irrelevant color on the target side. Under those conditions, we found hardly any difference between non-target and neutral color probes, speaking against a small GFBA effect to non-target colors and for some form of suppression of the magenta response. For the reviewer's inspection, we added a picture of the unpublished data below.

Unpublished data. Data from a similar unattended probe paradigm. Red, green, blue, and yellow were randomly assigned to subjects such that three of them were target colors and the fourth color served as neutral reference, never being task-relevant. A sliding-window t-test between 0-300m (sample-by-sample, 11.8ms window width) revealed no significant differences between non-target and neutral.

Figure changes: The results of the analyses of the magenta-colored probes are now provided as new Supplementary Figure S4.

Text changes: The respective analyses of magenta-colored probes are referred to in the Methods section of the main text (l. 732-733) and discussed in detail in the Supplementary.

- Participants covertly attended to a peripheral target while fixating a central cross. However, the authors did not use an eye tracker to monitor fixation. EOG was used to detect eye movement related artifacts in EEG/MEG epoch data. The authors should address if any trials were excluded from the main analyses because of larger EOG signal before the epoch window, which might indicate breaking fixation before the attention period.

Response: This is a good point. While EOG does not inform about the absolute eye position, it detects changes such as fixation breaks. Thus, as part of the artifact rejection (adjusting peak-to-peak amplitude measure thresholds) we removed all epochs where a larger EOG amplitude indicated horizontal or vertical eye movements during the analyzed epochs (including a 200ms pre-stimulus interval). That way we controlled for eye movements during the attention period as well as in the time period immediately preceding stimulation onset. We did, however, not specifically analyze the EOG signal between trials. To prevent participants from breaking fixation or directly looking at the stimuli and/or position of the upcoming

target, we took the following pre-cautions: Participants performed a couple of test trials until they felt comfortable solving the task without directly looking at the target stimulus. During the measurement, the experimenter closely monitored eye position using both video surveillance and the EOG signal. None of the participants showed excessive eye movements or repeatedly failed to maintain fixation.

Note, that neither breaking fixation in-between trials nor directly looking at the upcoming target should ultimately influence our main conclusions. Specifically, if participants would have looked directly at the target instead of fixating the cross, the probe in the opposite visual hemifield would still (and even more so) have been unattended. Of note, we actually ran a GFBA experiment where subjects directly looked at the target (presented centrally at fixation), while the probe was in the right visual field. The GFBA as elicited by the unattended probe could still be observed. Nevertheless, though GFBA operates across the whole visual field, it would have probably weakened the effect elicited by the unattended probe, if its retinotopic position would have differed among participants.

Text changes: We now added an extra paragraph to the EEG/MEG recording section where we detail the way eye movements were controlled. (l.795-805)

“Control of eye movements

To prevent participants from breaking fixation or directly looking at the stimuli and/or position of the upcoming target, we took the following pre-cautions: At the beginning of the experiment, participants were instructed to keep their eyes all the time on the central fixation cross (except for blinking pauses or breaks between blocks). They then performed a couple of test trials until they felt comfortable solving the task without directly looking at the target stimulus. During the measurement, the experimenter closely monitored eye position using video surveillance and the EOG signal. Eye movements that took place in the analyzed epochs (200ms before to 700ms after stimulus onset) were excluded based on the horizontal and vertical EOG (see below EEG/MEG - Epoching and Artifact rejection). None of the participants showed excessive eye movements or repeatedly failed to maintain fixation.”

Reviewer #2 (Remarks to the Author):

Bartsch and colleagues investigated the Global Feature-Based Attention (GFBA) effect whereby attending to a target stimulus of a particular color in one visual field results in enhanced processing of probes of that color in the opposite (unattended) visual field. This effect was manifested in previous studies by this group as an enhanced negativity at 150-250 ms in the event-related potential (ERP) recorded over the occipital scalp contralateral to the probe that matched the attended target color. The current study was designed to examine the processing of a probe having an alternative distractor color in the unattended visual field. Surprisingly, the distractor probe elicited a very early negative enhancement onsetting before 70 ms in relation to the ERP elicited by a “neutral” probe of a non-distractor color. This early negativity was enlarged on trials having faster RTs for discriminating the target color and on trials where the target color was repeated, and it was absent in a task where the same stimuli were discriminated for their spatial position rather than color. The authors interpret this early negativity, which was source localized initially to the dorsal lateral prefrontal cortex (DLPFC), as indicative of an inhibitory process that suppresses the competing distractor color information. It was suggested that the DLPFC initiates a command for an early “selection for rejection” process.

This study is well designed and methodologically rigorous. If the authors' interpretation of the early negativity is correct it would represent a remarkable new early inhibitory mechanism. I have some questions, however, about the timing of this early ERP modulation. Figure 4A shows that the initial feed-forward activation localized to area V1 begins at about 65 ms and is still small at 75 ms, whereas Figure 4B shows that the activation of the DLPFC, which also begins at about 65 ms, has nearly reached its maximum by 75 ms. So it seems that the frontal activation is nearly instantaneous with (or even prior to) the initial activation in primary visual cortex. How can this be?

Response: The reviewer points out that the activity in dIPFC appears very early in time relative to the initial feedforward response in V1, with both modulations seemingly arising instantaneously. We agree with the reviewer that this is remarkable. We would like to annotate, however, that the impression of an almost instantaneous onset of the feedforward response in V1 and the attention effect in dIPFC is to some extent an 'artifact' of the substantial difference in scale (source strength) between the source waves shown in panel A and B in Figure 4. To illustrate this, we have added a Figure in the Supplementary that plots the V1 (red) and IT response (green) of panel A with the dIPFC (yellow-dashed) and IT response (green-dashed) of panel B between 55 -100ms at the scale of panel B. To compensate for vertical noise offsets, all responses are set to 0 between 55-60 ms. Plotted this way, it becomes apparent that the dIPFC response arises with a delay relative to the V1 response.

Supplementary Figure S2. Direct comparison of stimulus elicited feedforward activity (Figure 4A) with the early distractor selection (Figure 4B) in the 60-100ms time range. The stimulus-elicited source waves in V1 and IT (red-solid, green-solid) are replotted together with dIPFC and IT source waves (yellow-dashed, green-dashed) underlying the very early distractor selection (DC minus non-target) at the scale of Figure 4B. To compensate for vertical noise offsets, all responses are set to 0 between 55-60 ms. As can be seen, the mere stimulus-elicited feedforward response (solid waveforms) is far stronger than that underlying the early distractor selection (DC minus non-target difference waveforms, dashed). When depicted at the same scale, it becomes obvious that the dIPFC activity arises after the onset of the V1 feedforward response but with a very short delay.

Figure changes: We realize that the limited comparability of the source waves in panel A and B of Figure 4 at small temporal differences is an issue and added for a better depiction Figure S2 to the Supplementary. It is now referred to in the main text in the discussion (l.568).

Still, despite the fact that the relative onset of responses in V1 and dIPFC is in the expected temporal order, the delay between V1 and dIPFC is, indeed, very short. An analysis of

response latencies across monkey cortical areas (Schmolesky et al., 1998, *J Neurophysiol*; Nowak et al., 1997, "The Timing of Information Transfer in the Visual System." In *Extrastriate Cortex in Primates*) revealed that the firing-response onset difference in frontal area FEF (50% response at 75 ms) and in V1 (66 ms) is very short (cf. Schmolesky et al., Fig2), and cannot be accounted for by a regular propagation of activity up-stream the cortical hierarchy. The authors suggest that cortical bypass connections from V1 via MT/MST to frontal areas must account for the minimal delay. Also, direct subcortical projections (e.g., via the MD or pulvinar, Barbas et al., 1991, *J Comp Neurol*; Goldman-Rakic and Porrino, 1985, *J Comp Neurol*) to frontal areas would be consistent with such fast response onset in dlPFC. Of course, these considerations on feedforward signal timing leave the question unanswered, how attentional processes can arise so fast in frontal cortex. However, work in the monkey suggests that frontal attention effects can indeed appear very early and consistent with the present observations. When monkeys performed a visual search task (Thompson and Schall, 1999, *Nature Neuroscience*), it was found that responses in FEF differentiated between hits/misses/false alarms already ~70ms after search frame onset. Furthermore, overtraining monkeys on one color in a color-based search task (Bichot, Schall and Thompson, 1996, *Nature*) caused color selectivity in FEF around 70-80ms after stimulus onset. This latency of color selectivity precedes the typical onset of such effects in extrastriate areas, but follows shortly the activity onset in V1 (ranging from 45-80ms). Those observations are well in line with the latency of DC-selection in dlPFC observed here. Specifically, participants were also effectively overtrained on the two possible target color combinations, as they were constant across the whole experiment. It would, of course, be interesting to see whether the early DC-effect would still appear when there was a greater variety of target colors and/or a more flexible target color assignment across the experiment (e.g., new color combination every block).

Of course, the dlPFC area seen here rather coincides with area VPA than with FEF. However, recent work (Bichot et al., 2015, *Neuron*) suggests that the training-dependent color selectivity in FEF is, in fact, inherited from area VPA, from where it is propagated to FEF (delay ~10ms), suggesting that color selectivity in VPA would arise at an even shorter latency than in FEF.

Text changes: We now added a detailed discussion of the early timing of the DC effect in light of the respective literature (discussion, l. 566- 581):

"As visible in Figure 4, the DC modulation in dlPFC (Figure 4B, yellow-dashed) follows the initial feedforward response in V1 (Figure 4A, red) with seemingly no delay. A higher-resolved comparison of the timing of the respective source waves (Supplementary Figure S2), however, shows that the dlPFC response arises with a small delay relative to the feedforward response in V1. Nonetheless, such small delay cannot be explained by a regular transmission of activity up-stream the cortical hierarchy. Response latencies in macaque frontal area FEF (50% response at 75 ms) were also found to arise with only minimal delay relative to V1^{59,60}, suggesting that cortical bypass connections or direct subcortical projections (e.g., via the mediodorsal nucleus or the pulvinar^{61,62}) play a role. Notably, when overtraining monkeys on one color in a color-based search task, color selectivity was seen in FEF already around 70-80ms after search frame onset⁶³. This selectivity follows shortly after the activity onset in V1 and precedes the typical onset of such effects in extrastriate cortex areas, consistent with the latency of the here observed DC-effect in dlPFC. Of course, the locus of source activity in dlPFC seen here rather coincides with the ventral prearcuate (VPA) region than with FEF.

However, recent work indicates that area VPA is the source of training-dependent color selectivity in FEF⁶⁴, suggesting that color selectivity in VPA would arise at an even shorter latency than in FEF.”

It would seem that initiation of a distractor suppression process would involve a sequence of events: (1) discrimination of the target color, (2) linking this target color with its alternative distractor color, a linkage that depends on memory and possibly inter-hemispheric information transfer and (3) initiation of a suppression of the distractor. How can all this processing possibly take place in just a few ms?

The point the reviewer raises here, is the most intriguing one. That the fast distractor selection is the outcome of a time-consuming sequence of decision processes is, indeed, hard to imagine. We would like to annotate that we do not think that the rapid frontal selectivity involves a sequential involvement of the three operations suggested by the reviewer. The effect rather arises as a consequence of a consistent preset-bias of the two target colors above all other probe-colors in the experiment. As in (Bichot, Schall and Thompson, 1996, *Nature*) subjects are overtrained on those colors and develop a strong template for the target colors, accelerating typical feature-based effects in FEF around 90-100ms (Bichot et al., 2015, *Neuron*) (search tasks) by 20 ms (Bichot, Schall and Thompson, 1996). We think the first and second step suggested by the reviewer may not be part of the initial response in frontal areas. Instead, the decision is based on a simple and time-efficient two-step process: Determine whether one or two target colors appear (no matter where they are presented). If two appear, highlight the color at the probe location for suppression. If just one target color appears, do not highlight (see schematic Figure two points below). That is, the dlPFC performs a simple decision among alternatives coded as opponent activations. Target detection by opponent coding has been documented in dlPFC (Kusunoki et al., 2010, *JoCN*), and has been proposed to represent a general mechanism of dynamic coding (Machens et al., 2005, *Science*). Its effect is a dramatic increase in the efficiency of coding. Note, at this early stage of selection, the subject may not even be aware of which target color is on the target side and which is on the probe side. Color identity plays no role at this point, and arises later during the GFBA phase of processing in extrastriate visual cortex. Furthermore, time-consuming callosal transfer may play no role. There is a bilateral VF representation in dlPFC, with cells preferring left and other cells preferring right VF targets (e.g., Lennert and Martinez-Trujillo, 2011, *Neuron*). A linking of the alternative target colors is therefore easy to accomplish without involving callosal transfer. Specifically, it has been proposed that the bilateral VF organization in dlPFC may facilitate competitive-interactions between items presented in opposite VFs via short-range connections. The preset bias in the present work may establish some form of opponent-bias for the alternative target colors appearing at the target and probe location. This would provide an efficient way to implement a rule resolving the ‘coding conflict’ on the fly. In fact, Bichot, Schall and Thompson (1996) suggest that the rapidity of the feature-based attention effects in FEF were due to plasticity that established a ‘habit’, i.e., an arbitrary stimulus-response association. It is likely that such habit-forming is at work in the present study, especially since the spatial layout of the stimulation display is fixed.

Text change: We now provide a detailed discussion of this issue, clarifying that the early DC selection is not supposed to be the result of a time-consuming sequence of preceding decision processes but that the rapid frontal selectivity rather resembles an implementation

of preset-biasing in the form of opponent coding (l. 582-606). A schematic depiction will be added to the Supplementary, see response two points below.

“Nonetheless, the early onset of the DC-modulation in dlPFC and its fast propagation to extrastriate areas is remarkable, given that a number of cognitive processes (discrimination of the PC, verify the DC, command suppression of the DC) would have to be put into operation in the short time range of ~10-20 ms. Though the present work cannot clarify the exact nature of the frontal control signal, it is unlikely that such fast distractor selection is the outcome of a time-consuming sequence of decision processes. Instead, we hypothesize that overtraining the two target color sets (used throughout the whole experiment) and the constant spatial layout (target always on the left) enabled subjects to establish a competitive link (opponent coding) between the alternative target colors in dlPFC that rapidly highlights whichever target color appears at the probe location. In fact, such competitive link between the color alternatives presented in the left and right visual field could easily be implemented in the dlPFC. Specifically, the dlPFC displays a bilateral VF representation, with some cells preferring left and other cells preferring right VF targets ^{e.g., 58}. It has been suggested that this bilateral VF organization may facilitate competitive interactions between items presented in opposite VFs (here: target and probe) via short-range connections, i.e., without the need for callosal transfer. Moreover, target detection by opponent coding has been documented in dlPFC ⁶⁵, and has been proposed to represent a general efficient mechanism of dynamic coding ⁶⁶. Hence, the dlPFC may establish a competitive link between the target color units preferring the left (target) or right VF (probe). This would allow for a fast two-step binary decision process that can operate entirely on the feedforward response of color units in the dlPFC: (1) Determine whether or not both target colors appear on the screen (irrespective of their location), (2) If yes, label the color that is reported by units preferring the probe’s visual field (see Supplementary Figure S3 for a schematic illustration). Note, at this early stage of selection, color identity plays no role, the competition process just highlights the target color, that happens to appear with the probe even when subjects are not aware which target color is on the target side and which is on the probe side.”

The transmission time from V1 to the DLPFC also needs to be taken into account. The authors need to explain how this selection for rejection could work within the time frame of the source waveforms shown in Fig 4.

See response to the first point. We now provide a better depiction of the temporal order of the source waves and added experimental work showing that the transmission time of feed-forward activity from V1 to frontal areas can be very short.

Looking at the literature, it seems that overt attention to a specific color generally modulates the ERP starting at around 100 ms (e.g., Anllo-Vento et al. Human Brain Mapping 1998; Schoenfeld et al. Cerebral Cortex 2007). The earliest modulation I could find was found by Zhang & Luck (2009), who found that an attended-color probe elicited an enhanced positivity onsetting at 80 ms. In all these studies a particular color was attended top-down, so that color-specific feed-forward pathways could be primed before stimulus delivery, and therefore a very early modulation of the feed-forward sweep was possible. In the present study, however, it is not possible to attend to a specific color but rather to discriminate between two alternative colors. How is it possible for such discrimination plus the additional steps listed above to be achieved faster than simple top-

down feed-forward biasing? The authors need to explain further how such a proposed distractor suppression mechanism could possibly work within the observed time frame.

One critical issue the reviewer raises here is whether two colors could be attended simultaneously, or put differently, whether it is possible to establish a preset bias for two colors before stimulus onset. This has been experimentally verified by showing that multiple control settings can be implemented to guide search by two colors simultaneously (Irons, Folk, and Remington, 2012, JEPHPP; Stroud et al., 2012, JEPHPP; Grubert and Eimer, 2016, JEPHPP, Roper and Vecera, 2012, Psychon Bull Rev).

The three studies cited by the reviewer differ in one very important aspect from the present color task. In all those studies, color served to define the target, but color identity was never the to-be-discriminated target information (luminance change in Zhang & Luck 2009, Anllo-Vento et al., 1998, a mere color change in Schoenfeld et al. 2007). That is, those studies do not involve a conflict of coding color identity in the moment of target discrimination. As outlined above and illustrated in the Figure below, we believe that the early DC-modulation does not reflect the discrimination of the colors in the sense of building a conscious representation of their identity. Instead, based on an overall preset bias for both colors (red and green), the dIPFC established a learned competitive link between the target color units independent of whether they prefer the left (L) or right VF (R). This implements a binary decision network operating on the feedforward response of color units in dIPFC as shown in the Figure below. It returns a “1” when only one target color is present (red, non-target probe, PC probe), but a “0” when both target colors appear (red and green, DC-probe). If “0” is reported, the only additional operation required is to highlight the color reported by the RVF preferring units (here: green). We have added an in-depth discussion of the issue in the discussion section. In addition, we provide a schematic illustration of the competitive coding account in the Supplementary materials.

Supplementary Figure S3. Schematic illustration of a potential mechanism to facilitate rapid DC selection at the probe location. (A) Let the dIPFC contain units with learned color-selectivity (here: red and green), with some preferring left visual field (VF) targets (L) and some preferring right VF targets (R). Among those implement a competitive link (opponent coding) between red and green units ($|red - minus - green|$). **(B)** Different states of competition signalling the presence of one or two target colors exemplified for trials with a red target in the left VF (Left-red units always active, filled red) and a probe in the right VF. Exemplary stimulus displays underlying the different competition states are shown on the left. For a **non-target probe**, only the target-red unit is active with the competition reporting 1. For a **PC probe**, the left- and right-preferring red units are active, but no green unit is active, the competition reports 1. For a **DC probe**, left-preferring red and right-preferring

green units are active, with the competition reporting 0. When the competition is 0, highlight the color reported by the active units preferring the right VF (step 2). Importantly, as the competitive link is highly trained, the decision process can operate entirely on the feedforward response of color units to accomplish a rapid DC selection at the location of the probe. Note, the decision process is time-efficient as it does not require to access the identity of the colors presented at the target and probe side.

Figure changes: We added a schematic illustration of the competitive coding account as Supplementary Figure S3.

It would be of interest to examine the ERPs recorded over the hemisphere contralateral to the targets on exactly the same trials where the early negativity was revealed contralateral to the distractors. Surely the authors have looked at this. I assume they would predict no early negativity in the appropriate difference waves. If there were such an early modulation it may require some re-thinking of the suppression hypothesis.

The reviewer asks an important question. In fact, our experiment was designed in a way that one would not expect any differences contralateral to the target when comparing probe color conditions. Specifically, any comparison between attended (DC, PC) or unattended (non-target) probe colors contain identical attention conditions for the target side (“attended color, PC”). Nonetheless, if there was such an early selection not only for the distracting color probe but also for the target, this would, indeed, call into question the function of the reported early modulation in terms of specifically highlighting/suppressing the distracting color. To address this issue, we analyzed the ERPs contralateral to the target on both DC and PC trials. A sliding t-test (0-300ms, window 11.8ms) revealed significant responses contra target (PO4/PO8) only fairly late in time (DC trials: 242-300ms; PC trials: 218-246ms). As can be seen in the topographical field maps, there is no prominent response maximum contra target in the early (90ms) or late (190ms) time range. On DC trials, the response at PO4/PO8 peaks at 270ms, showing a clear topographical distribution maximum contra target. This late modulation might hint at a delayed target processing on DC trials. For PC trials, the significant modulation contra target seen in the ERP waveform around 230ms reflects a spill of activity from the strong contra probe modulation in this time range (cf. field distribution at 230ms). Notably, there is no significant negativity contra target in the early time range neither on DC nor PC trials. Hence, the early effect is exclusively found for the probe response, i.e., the distracting color. The respective analyses are now added as Supplementary Figure S5.

Figure changes: Contra target analysis added as Supplementary Figure S5.

Text changes: Reference to Figure S5 in Methods (l. 739-743): “Note that any comparison between attended (DC, PC) and unattended (non-target) probe colors involves identical attention conditions at the target side (target always drawn in PC). Hence, attention-related response differences are expected to largely cancel contralateral to the target (see Supplementary Figure S5 for ERPs recorded contralateral to the target).”

Figure S5. ERP results for the color task: DC/PC minus non-target differences contra target. (A) Shown are the ERP waveforms elicited by the target at PO4/PO8 (signal averaged, purple line) for the DC minus non-target (upper row) and PC minus non-target difference (lower row). Rectangles highlight time ranges of significant brain response variations contra probe as derived by the 2x3 rANOVA in the main analysis. For a better comparison, the ERP waveforms contra target are shown together with the respective ERPs contra probe (PO3/PO7, signal averaged, thin dashed grey or black line) replotted from Figure 3B. Contra target, a sliding window t -test (sample-by-sample, 11.8ms window) between 0-300ms revealed only significant modulations ($p < 0.02$ for more than five consecutive time samples, see Methods) in late time ranges (purple horizontal bars, DC: 242-300ms, PC: 218-246ms). **(B)** The respective topographical field maps on the right display representative time points at either DC or PC modulation maxima, positions of electrodes used for the analyses are highlighted (dashed grey or black and purple ellipses). On DC trials (upper row), there is a prominent response maximum contra target at 270ms, which might reflect a delayed target processing. For PC trials (lower row), the significant modulation contra target seen in the ERP waveform around 230ms represents a spill of activity from the strong contra probe modulation (cf. field distribution at 230ms). Notably, there is no significant negativity contra target in the early time range neither on DC nor PC trials. Hence, the early biasing exclusively appears on DC trials contra probe, i.e., for the distracting alternative target color.

A few minor wording improvements:

--Line 344: change "would require" to "would be required"

We rephrased the sentence. It now reads: "When one would strengthen the cortical representation of the PC, a larger early modulation for the DC would be required to instantiate a prioritized representation of the DC against this stronger PC bias." (now line 370-372)

--Line 362: change "enforces" to "reinforces"

We changed the wording accordingly. (now l. 387)

--Line 374 "...which was much smaller...." Changed accordingly. (now l. 399)

--Line 759: "...relative to the...." Thanks. (now l. 873)

Reviewer #3 (Remarks to the Author):

The current study assessed whether feature-based enhancement of target colors and suppression of distractor colors. Participants performed a task where they classified the orientation of a circle and tried to ignore a distractor in the opposing hemifield. The distractor could be the presented target color, a distractor color, or a neutral color. EEG and MEG were measured in response to these stimuli. Three experiments seemed to indicate that the representation distractor was initially enhanced before it was suppressed.

The results are interesting and address an important question. However, I have some reservations about the interpretation of the ERP waveform. Namely, one could argue that the early negativity toward the distracting color as indicates attentional suppression, not attentional enhancement of the distractor.

Major Comments:

1. The authors argue for a “selection for rejection” based upon an early negative enhancement in the ERP waveform, which they propose reflects attentional enhancement of the distractor probe. However, many previous studies have used P1 enhancement effects as an indicator of feature-based attention (for reviews, see Hillyard et al., 1998; Mangun, 1995). The basic pattern of results in these studies is that the P1 wave is larger (i.e., more positive) for attended items than unattended items (Heinze et al., 1994). As shown in Figure 3, the distractor colored probes (DC) produce a smaller P1 than the non-target baseline. In my view, this relative negativity seems more consistent with the interpretation that attentional allocation was reduced for the distractor color (DC) probes compared to baseline. In other words, could argue that the distractor color was initially suppressed, not attended as the authors have claimed. At a minimum, some clarification of the current results in the context of P1 enhancement is needed. But ultimately a strong reinterpretation of the data seems warranted.

Response: We thank the reviewer for raising this important point. It is true, that the mere polarity of a voltage deflection is ambiguous as to whether it reflects neural enhancement or suppression. The local field potential (LFP) variation underlying a positive or negative ERP deflection can be caused by an enhancement or attenuation of excitatory or inhibitory activity. Moreover, the same LFP can appear as positive or negative polarity deflection depending on the field projection of the underlying current dipole to the recording electrode. Critically, the polarity modulation must be interpreted relative to a neutral (baseline) activation serving as a reference.

The reviewer suggests an alternative interpretation of the DC effect in terms of a smaller P1 response to DC as opposed to an enhanced negativity, by referring to work that documented larger P1 responses to index an increased allocation of attention. This is indeed a possible interpretation that cannot be ruled out entirely, and we now discuss the possibility of a P1 attenuation in the revised manuscript (see point 3 below). However, this interpretation seems less likely than the one we pursued in the manuscript for three reasons detailed below (point 1-3).

Finally, we completely agree with the reviewer that, regarding the neuronal activity underlying the ERP response, we cannot ultimately decide as to whether the relative negativity results from a neural enhancement or suppression/attenuation. We accordingly changed the manuscript at relevant points to entertain an interpretation of the early DC

modulation in terms of both, neural enhancement and attenuation. We would like to annotate that the general conclusion of a prioritized DC processing – in terms of enhancement or attenuation – does not change.

The possibility of a P1 attenuation accounting for the early DC modulation is less likely for the following reasons:

(1) The interpretation in terms of a P1 would imply that the PC elicits a larger P1. But then, we must assume that the non-target color shows the same P1 enhancement as the PC (P1 amplitude of non-target similar to that of PC). This is hard to reconcile with the cited work on the P1 component, as it would mean that the non-target color is as attended as the target color, although it never appears on the target side. To rectify this, one could assume that both the PC and non-target color elicit no response. But then, the smaller positivity elicited by the DC is, in effect, a larger negativity, which corresponds with the interpretation offered in the manuscript. We agree with the reviewer that the interpretation of the relative polarity effect is inherently ambiguous, and that the DC-effect could reflect a prioritized attenuation of the DC. However, this would not change the interpretation of the present observations. The DC-effect could reflect an increased response of inhibitory neurons and would still be a prioritized neural signature of the DC. We have extended the discussion section to address this issue (l. 527-548). We now clearly indicate that the nature of the neural process underlying the DC effect cannot be determined, so that we also offer the alternative interpretation of the reviewer.

(2) The interpretation of the DC-effect as reduced allocation of attention would be incompatible with the interpretation of the subsequent GFBA effects. For the latter, the smaller response to DC versus PC is taken to indicate a reduced allocation of feature-based attention – an interpretation that is commonly agreed on, and which is experimentally verified. If we assume that the increased DC-negativity reflects attenuation, the following GFBA reduction of the DC would then imply enhanced allocation of feature-based attention to the DC. While this interpretation cannot be ruled out, it causes complications of the interpretation that would run against a substantial body of experimental data.

(3) The reviewer refers to seminal work by Hillyard et al. (1998), Mangun (1995), and Heinze et al. (1994). It is important to acknowledge that the P1 enhancement documented in this work is not an index of feature-based attention. Instead the P1 enhancement is an index of spatial orienting of attention (typically investigated with Posner-type cuing tasks). There is common agreement that the P1 reflects the neural amplification of stimuli appearing at an attended versus unattended location. That is, a greater P1 amplitude indexes stronger spatial focusing. The feature-based discrimination of the target was reflected by the subsequent N1 component (Mangun & Hillyard, 1991). In the present experiment, the effect of spatial focusing is controlled. That is, PC, DC, and non-target trials are identical in this respect as spatial attention was equally focused on the target in the LVF and equally withdrawn from the probe in the RVF. Hence, we would not expect to find any modulation of the P1 to the probe when comparing among our experimental conditions.

Nonetheless, there is, indeed, a study reporting a positivity in the P1 time range as an index of feature-based attention (Zhang and Luck (2009)). This study used a very different experimental design, involving sustained focusing onto a mixture of flickering colored dots in one VF, with subjects being required to attend one color-defined dot group and ignore the other group. As in the present study, color probes were presented in the other VF. Attended-color probes elicited a positive enhancement between 100-150ms, relative to nontarget-color

probes. Theoretically, the early DC modulation may reflect a reduction of such positive enhancement. However, despite the limited comparability between experimental designs, this observation is incompatible with the present data, because we find no difference between the PC and the non-target color. Moreover, the P1 effect of Z&L disappeared when presenting target and distractor dots separately on the target side (to eliminate color competition within the focus of attention), which is a situation more close to the present experiment. Still, we think the reviewer raised an important issue that warrants a discussion of our findings in light of the feature-P1 modulation found by Z&L, which is now added to the discussion section (l. 527-539).

Text changes: We now added a paragraph in the discussion to clarify this issue. Moreover, we refined the wording carefully throughout the manuscript (especially removing the term “negative enhancement” for the early modulation and entertaining attenuation as an alternative explanation).

l. 300-301: “Alternatively, the early DC modulation may reflect an immediate attenuation to facilitate later rejection.”

l.459-460:” This selection process, which may represent a rapid enhancement or attenuation of neural activity coding the DC,...”

l. 527-539: “One possible interpretation worth considering is that the early DC-related modulation may represent an attenuation of a positive ERP deflection rather than an enhanced negativity. In fact, under certain experimental conditions, it was shown by Zhang and Luck that GFBA can influence the feed-forward sweep of processing in visual cortex very early on, in form of a positive-going modulation in the P1 time-range⁴⁰. Theoretically, the early DC modulation may reflect a reduction of such P1 response. However, the feature-based P1 modulation in Zhang and Luck was only seen under conditions of color-competition in the spatial focus of attention, but not when a single color was presented (experiment 2 in⁴⁰). As the present experiment did not involve color competition in the focus of attention (a single color appeared in the focus of attention), such P1 effect would not be expected to appear. Furthermore, an interpretation in terms of an attenuated P1 response for the DC relative to the PC would have to reconcile with the fact that the PC does not differ from the non-target color in this early time range. Hence, while the possibility of an attenuated P1 response cannot be ruled out entirely, it remains a less likely interpretation.”

l. 540-548: “It should also be pointed out that the neural process underlying the prioritized selection of the DC cannot be clarified with the present data. We assume that the early negative polarity modulation represents enhanced neural activity of units coding for the DC, which highlights the representation of the color for subsequent attenuation. It is alternatively possible that the early negativity reflects activity of inhibitory units instantly attenuating the neural representation of the DC. This diminished representation may then be the basis of the subsequent attenuation in the GFBA time range. In any case, whether the early DC-related modulation is of excitatory or inhibitory nature, it represents a temporal priority signal that highlights the DC for subsequent attenuation.”

Examples for word changes:

- l. 45: “neural gain” -> “neural modulation”
- l. 220/293: “bias” -> “effect”/“modulation”
- l. 233: “enhanced negativity” -> “higher relative negativity”
- l. 215: “negative enhancement” -> negative deflection
- l. 242: “enhancements” -> “amplitude modulations”
- l. 425: “negative enhancement” -> “modulation”

2. The above P1 reinterpretation could also be applied to the other results. For example, in Figure 5, this new interpretation would suggest that successful early suppression of the distractor color (DC) probe resulted in a faster the RTs to detect the target stimulus (Figure 5). In other words, if the negativity in the P1 range indicates suppression (rather than attentional allocation), then the fast RT trials would be the trials with successful early suppression of the distractor color.

As outlined above, the P1 interpretation is possible but unlikely, as it complicates the overall interpretation of the GFBA effects. However, we agree with the possibility that the negativity in the P1 range reflects a signal that is inhibitory in nature (enhanced response of inhibitory neurons). This alternative interpretation is now entertained in the manuscript and discussed in detail (l. 527-539), see answer to previous point.

3. Again, if you assume that relative negativity (compared to baseline) in the P1 range reflects suppression (i.e., less P1 enhancement), Figure 6 would suggest that repeating the target color causes the DC to be suppressed not enhanced.

We entirely agree, under the alternative interpretation suggested by the reviewer, the increased relative negativity would reflect DC suppression. As already mentioned, we thank the reviewer very much for pointing this out and we now entertain this alternative interpretation in the manuscript. But with this interpretation, the issue of there being no difference between the PC and the non-target color remains. In other words, the relative DC negativity represents a distinct modulation signaling the presence of the DC at the probe side. Insofar, the discussion of the DC modulation does require generalization, which we now provide in the manuscript, but it does not change the general interpretation of the DC modulation indexing a process that highlights (via enhancement or suppression) the presence of the DC with temporal priority.

Minor Comments

Figure 5: For the early vs. late analysis, might it be useful to show the actual difference waveforms for fast vs. slow RT (instead of just the bar plots). If they are extremely noisy, the authors might consider including them as a supplement.

Response: We agree that it would be informative to show the actual difference waveforms and added them now to Figure 5B.

Figure 5. Median response time split analysis. (A) Difference waveforms for the PC (black) and DC (grey) replotted together from Figure 3B for better comparison. Rectangles indicate previously determined early and late time ranges of significant experimental variation. (B) Median split into fast (upper row) and slow (lower row) response times (RT) for DC (grey) and PC (black). Left column: fast (upper row) and slow (lower row) minus non-target difference waveforms. Stars indicate significant mean amplitude modulations in the early and late time ranges ($p < 0.05$). An explorative sample-by-sample sliding t-test (0-100ms, 11.8ms window) found no effect for fast PC trials within or before the early time window. Right column: Mean GFBA amplitudes of the early and late time range are shown for fast (upper row) and slow (lower row) responses. For DC, the early negativity was higher when participants responded fast compared to slow, which was inversely correlated to the size of the late bias (significant early/late GFBA amplitude \times fast/slow RT interaction, $p = 0.00035$, see text for details). For the PC, in contrast, there was no significant difference in GFBA amplitudes between fast and slow responses but always a strong late bias. (C) 3D current source density map for the early DC biasing (fast RT trials minus slow RT trials) between 70-95ms (25ms average). As can be seen, differences in source activity for fast and slow DC trials emerge in posterior extrastriate visual cortex (around V3).

Figure changes: The difference waveforms were added to Figure 5B. Note that we also adapted the Figure layout to match that of Figure 6B. Responses of PC and DC are now shown together in the same coordinate system (fast responses upper row, slow responses lower row), bar graphs changed accordingly.

Text changes: The difference waveforms are now described in the results section (l.312-319): “The DC minus non-target difference waveform shows a significant early modulation for fast ($p=0.0019$, grey line in Figure 5B upper row) but not for slow ($p=0.3755$) DC trials (grey line in Figure 5B lower row). In contrast, in this early time range, no response difference is seen between fast and slow PC trials (upper and lower black traces). Furthermore, while there is a small GFBA response to fast DC trials in the late time range which onsets with the GFBA response to fast PC trials, it is rapidly attenuated. This is in contrast to slow trials (Figure 5B lower row), where the late GFBA response to DC trials is not attenuated. Note, the early modulation in DC trials is present in fast but effectively absent in slow trials.”

p. 14 When introducing the idea of intertrial color priming, it seems important to cite the seminal studies of Maljkovic and Nakayama (1994, 1996).

Response: Thanks for mentioning this, we now cite the studies of Maljkovic and Nakayama about color priming (l. 375-376).

p. 15 Perhaps the “selection for rejection” mechanism being proposed is also similar the “search and destroy” models proposed by Egeth and colleagues.

Response: We thank the reviewer for annotating this point. We now embedded the “search and destroy” model in the discussion.

Text change: l. 625-628: “These observations were discussed in terms of prioritized selection for rejection of the distractor singleton, which dovetails with psychophysical work suggesting that the inhibition/deprioritization of distractors in visual search involves the prior attentional selection of those items (attentional ‘white bear phenomenon’) ^{69,70} akin to a search and destroy process ⁶⁹.”

Reviewers' Comments:

Reviewer #1:

Remarks to the Author:

Thank you to the authors for their clear and thorough reply. My opinion of the original version of this manuscript was very positive. I had only a couple general issues and a handful of specific/minor issues, all of which were addressed to my satisfaction in the revised version. I also read the comments from the other reviewers and believe that the authors did an excellent job of responding to their concerns. In my opinion, this paper makes a good contribution to the literature and is worthy of publication.

Reviewer #2:

Remarks to the Author:

The authors have provided detailed responses to the reviewers' critiques and have made extensive revisions to the manuscript to accommodate them. The changes made in response to my comments are by and large convincing.

In particular, the authors have cited data from monkey experiments that may account for the extremely short latencies of the ERPs elicited by the distractors. They have also proposed a pre-set binary decision mechanism that may explain how a distractor suppression process could be implemented so quickly. Surely other explanations for this early ERP effect are possible, but overall I find this to be a provocative study of a remarkable phenomenon that calls for further investigation.

Reviewer #3:

Remarks to the Author:

The authors have adequately addressed my previous concerns in the current manuscript.